# Evaluation and Spatiotemporal Differentiation of Cultural Tourism Development Potential: The Case of the Middle and Lower Reaches of the Yellow River

Yuying Chen [1,2,3], Yajie Li [2], Xiangfeng Gu [4], Qing Yuan [1,2,5], Nan Chen [1,2,3] and Qi Jin [1,2,*]

[1] Research Institute of Study Travel, Henan University, Kaifeng 475001, China; yychndx@henu.edu.cn (Y.C.); yuanqing@henu.edu.cn (Q.Y.); 10020053@vip.henu.edu.cn (N.C.)
[2] School of Cultural & Tourism, Henan University, Kaifeng 475001, China; liya@henu.edu.cn
[3] Key Research Institute of Yellow River Civilization and Sustainable Development Collaborative Innovation Center on Yellow River Civilization/Collaborative Innovation Center of Yellow River Civilization Provincial Co-Construction, Henan University, Kaifeng 475004, China
[4] Department of Geography, The University of Hong Kong, Hong Kong 999077, China; u3010642@connect.hku.hk
[5] College of Geography and Environmental Science, Henan University, Kaifeng 475004, China
* Correspondence: 10020085@vip.henu.edu.cn; Tel.: +86-185-3946-6548

**Abstract:** Cultural tourism development potential (CTDP) is the future value and supporting force of the environmental value, economic and social efficiency, innovation ability and supporting system of cultural tourism. At present, there are few relevant studies on CTDP, but the research results on the tourism development potential of cultural heritage are relatively rich, and the existing evaluation methods lack comprehensiveness, dynamics and visualization. Based on systems theory and sustainable development theory, this article attempts to innovate and collect time series data through the entropy method, multi-index comprehensive evaluation method, spatial kernel density estimation method, and centroid transferring model. The temporal and spatial evolution characteristics and the CTDP of 43 cities in the middle and lower reaches of the Yellow River are examined and analyzed. It is found that the CTDP in the middle and lower reaches of the Yellow River is divided into five levels; the overall potential intensity of the research area is small and has significant spatial differences; influenced by the time factor, the interaction and spatial correlation of within the research area are significant; the development of regional cultural tourism has strong regional dependence in the short range. The center of potential gradually moves to the geometric center. This study is significant for promoting the sustainable development of economic tourism in cradles of world civilization.

**Keywords:** cultural tourism development potential (CTDP); potential levels; spatiotemporal differentiation; urban tourism; the middle and lower reaches of the Yellow River

## 1. Introduction

The Nile, Euphrates, Tigris, Indus and Yellow Rivers are the birthplaces of world civilization throughout history. The Yellow River gave birth to the Chinese civilization, and retained the ancient capital's civilization, ancient science and technology, political system, agricultural and commercial civilization, literary classics and other rich cultural tourism resources. In this new era, ecological protection and high-quality development of the Yellow River Basin has become China's national development strategy. The "Outline of the Yellow River Basin's Ecological Protection and High-quality Development Plan" issued by the Central People's Government of the People's Republic of China clearly points out from a strategic perspective that the Yellow River culture should be protected, inherited and promoted, and the philosophical thoughts, humanistic spirit, values and ethics contained in it should be deeply discovered. Diverse and harmonious Yellow River cultural demonstration zones, such as Hehuang-ZangQiang, Guanzhong, Heluo and Sanjin,

should be constructed, and a Yellow River cultural tourism belt with international influence should be established.

As early as the 16th century, the Grand Tour put a premium on exotic cultural experiences [1]. In 1999, UNWTO published the "Global Code of Ethics for Tourism", which emphasizes that tourism should be developed under the premise of protecting culture [2]. Since the 21st century, the tourism development and sustainable development of cultural heritage has attracted much attention from scholars, which has triggered a discussion on the relationship between culture and tourism in academia, and it is believed that there is both partnership and synergistic symbiosis between culture and tourism [3–6]. The study of cultural tourism began in the United States in the fifties of the last century; scholars believe that cultural tourism aims to meet the cultural needs of tourists and cause tourists to think about the social and human aspects of tourism activities, including handicrafts, art, music, architecture, perception of tourist destinations, monuments, festivals, heritage resources, technology, religion, education and other types of content [7]. At present, the research on cultural tourism mainly focuses on the cultural tourism experience, cultural tourism authenticity and the tourism development of cultural heritage [8,9]. There are some studies on the tourism development potential of cultural heritage, and there are almost no studies on CTDP, but cultural significance has increasingly become an important dimension in assessing regional tourism development potential [4].

In 2009, the Ministry of Culture of China and the China National Tourism Administration issued the "Guideline on Promoting the Combined Development of Cultural Tourism", proposing to promote tourism consumption with culture and to promote the integration of cultural and tourism industries. In the 2021–2025 period, one of the guiding principles of China's tourism development is to "promote the deep integration of culture and tourism", that is, to practice the coordinated and symbiotic development of cultural tourism with quality at multiple levels of resources, products, industries, markets and technologies. In practice, the establishment of Yellow River National Cultural Park provides a good environment for the development of Yellow River cultural tourism, and Yellow River cultural tourism development has entered a critical period. What would be its future development potential? Are there regional differences? How to inherit the Yellow River Civilization? The discussion of this series of theoretical questions is of great significance and is conducive to the high-quality sustainable development of Yellow River cultural tourism.

Development potential measurement is the important basis and key step for regional sustainable development, and regional or national tourism development potential has always been the focus of tourism research [10,11]. The existing literature on tourism development potential mainly reveals the univariate potential model and multivariate descriptive evaluation, and rarely conducts research and judgment on the regional dynamic change process and development trend of CTDP [12–14]. Based on comprehensive potential assessment, this paper combines geographic information systems (GISs) for in-depth analysis. It mainly uses cluster analysis, spatial kernel density estimation, centroid transfer model and Kriging interpolation analysis.

With the rapid development of cultural tourism, its contribution to regional development is increasing, and the competition between regions is becoming increasingly fierce. CTDP has increasingly become a field of concern for scholars. This paper will integrate the ideas of technical evaluation and comprehensive evaluation and construct the evaluation index system of CTDP, based on combing the existing literature on CTDP [13,15–20] and combining the essential characteristics of cultural tourism. Using the relevant data of 43 cities in the middle and lower reaches of the Yellow River from 2009 to 2020 for weight determination, an evaluation model is established to classify potential levels and reveal spatiotemporal differences. From the perspective of potential, the evaluation index system of CTDP enriches the research on cultural tourism and regional sustainable development and provides a decision-making reference for the high-quality and sustainable development of cultural tourism in the basin where world civilization originated.

## 2. Materials and Methods

*2.1. Research Area*

This paper takes the middle and lower reaches of the Yellow River as a case to study its CTDP. The Yellow River basin is the birthplace of Chinese civilization, comprehensively recording the origin, formation and development process of Chinese civilization and giving birth to farming civilization, urban civilization, ancient Chinese characters and bronzeware, social differentiation and ritual construction, political system, and cultural symbols such as Confucianism, Taoism and commerce. These have made great contributions to the formation of the Chinese national community and are of great significance to the sustainable development of the world's four ancient civilizations. The middle and lower reaches of the Yellow River are located in central and eastern China, between the North China Plain and the Jianghuai region. The terrains are mainly plateaus, plains and mountains, covering an area of 550,800 km$^2$, covering 43 cities in five provinces of Gansu, Shanxi, Shaanxi, Henan and Shandong [21], as shown in Figure 1. The middle and lower reaches of the Yellow River boast ancient Chinese capitals such as Xi'an, Luoyang, Kaifeng and Anyang, as well as rich cultural tourism resources such as ancient cities, monuments, ruins and cultural relics. According to statistics from the five provinces in the middle and lower reaches of the Yellow River, cultural tourism development in the research area has become a major sector of the regional economy since 2019, with its total revenue reaching 3716.948 billion RMB, accounting for 21.02% of the region's GDP.

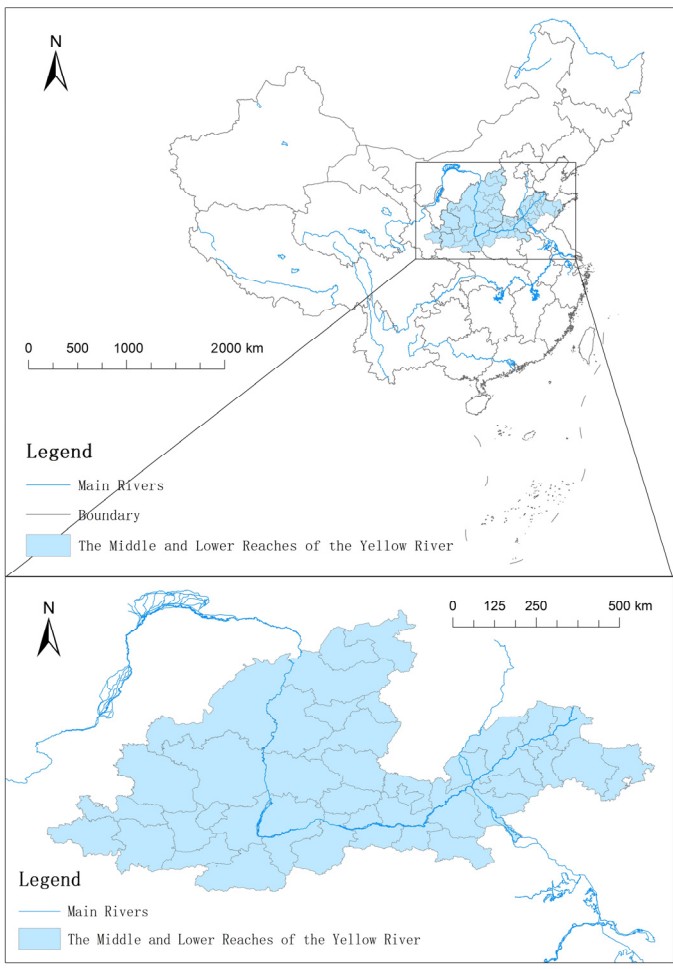

**Figure 1.** Spatial scope of the research area.

## 2.2. Literature Review

### 2.2.1. Cultural Tourism Development Potential

Development potential refers to the value and innovation capability of a region's supporting system under the premise of sustainable development in a certain period in the future [22]. CTDP can be understood as the development value and socioeconomic efficiency of cultural tourism resources, the innovation ability of cultural tourism development and the value and supporting force of its supporting system in a region on the basis of maintaining the sustainable development of cultural tourism in the future. In general, the research results of the evaluation of the tourism development potential of cultural heritage are relatively rich, and the research objects are diversified, involving world heritage sites, linear cultural heritage, agricultural cultural heritage, intangible cultural heritage, etc. [23–27], but there are few research results on CTDP. In the tourism development of cultural heritage, it is very important to evaluate the value and development status of cultural heritage resources from two dimensions [13,28,29]. Intangible cultural heritage integrates the authenticity of intangible cultural heritage, inheritance significance and the inheritors, and it plays a positive role in promoting the positive symbiotic relationship between tourism resources and tourism development potential and can promote the sustainable development of cultural heritage tourism [30].

In general, the evaluation index system that can be used for reference in CTDP evaluation is mainly constructed from the influencing factors and market players. From the analysis of the influencing factors of development potential, scholars pay attention to its internal and external factors, that is, the self-development potential of the tourism industry (internal resources and benefits) and the external guarantee and support (infrastructure, services and environment) [31–33]. From the perspective of tourism market players, scholars construct an index system from the market demand and the supply and demand side of tourism factors (attraction and cultural tourism enterprises) [13,34–36]. In terms of the evaluation indicators of CTDP, scholars mainly use indicators such as cultural tourism consumption capacity, operating accommodation capacity, number of tourists, number of overnight stays, average stay time and public and private sector investment in cultural tourism projects [37–39] to evaluate CTDP from multiple perspectives such as regional tourism planning, heritage protection strategy formulation and the cultural tourism industry.

At present, the existing research on cultural tourism development potential is mainly reflected in three aspects. First, although cultural heritage is a public commodity, it is the tourism development potential that can promote the sustainable development of regional tourism. Therefore, the tourism development potential of cultural heritage should be fully explored [40,41]. Second, cultural significance has increasingly become an important dimension in the evaluation of regional tourism development potential, which is composed of indicators such as cultural value, heritage protection, product value and experience value [42–44]. Third, the evaluation indicators begin to be diversified, but there are more univariate and binary (not multivariate) index evaluations, while the evaluation index system is relatively rare [13,14,18,19].

### 2.2.2. Development of the Yellow River Cultural Tourism

Cultural tourism has become an important research field for regional social and economic development and may become one of the trends of future tourism research [45]. Currently, research on cultural tourism mainly focuses on the cultural tourism experience and its authenticity, the sustainable development of cultural tourism, the integration of cultural tourism, the utilization of cultural tourism resources, and the development of the cultural tourism industry [8,9]. The proposal and implementation of the dual national strategies of the Yellow River Cultural Tourism Belt and Yellow River National Cultural Park have drawn much attention to the research on the development of Yellow River cultural tourism. The existing literature mainly focuses on new cultural tourism formats and the tourism path to inherit and carry forward the Yellow River culture. From the perspective

of industry, the development efficiency of tourism in the middle and lower reaches of the Yellow River is at a medium level [46], but there is a positive correlation between the dual-cycle industrial chain of culture and tourism in the Yellow River basin and regional economic development. Tourism products with Yellow River culture as the core, such as the Yellow River tour, Silk Road tour, Three Kingdoms tour, Confucian and Taoist cultural tour, etc., have led to the diversification of new cultural tourism formats [47–51]. From the perspective of the inheritance and promotion of Yellow River culture, the development of cultural tourism is the main approach to organizing, presenting and spreading the story of the Yellow River. The key to telling the story of the Yellow River well in the development of tourism is to effectively transform the multi-dimensional space of cultural tourism into a narrative space on different levels. By virtue of the high adaptability of traditional villages in the Yellow River Basin to develop tourism, the construction of Yellow River cultural tourism villages can be achieved. It can also fully explore the cultural tourism function of intangible cultural heritage in efficiently inheriting and promoting Yellow River civilization [52–54]. On the whole, there is more descriptive literature on the development of Yellow River cultural tourism, fewer quantitative studies, more on development status and strategies and less on the development potential and trends.

### 2.3. Data Sources and Standardized Processing

The data in this study are mainly from statistical bulletins and statistical yearbooks issued by the Chinese government at all levels during 2010–2021, as shown in Appendix A. The relevant data of the cultural tourism development potential of 43 prefecture-level cities in the research area during 2009–2020 were retrieved and supplemented by smooth interpolation for a few missing data. The range method is used to standardize the original data and eliminate the dimensional differences.

### 2.4. Research Methods

#### 2.4.1. Entropy Method

The smaller the information entropy of an index, the greater the degree of variation in the index value, the more information it provides, and the greater the role it can play in a comprehensive evaluation, and vice versa [55]. The calculation process is as follows:

Firstly, the entropy value ($e_i$) and difference coefficient ($d_i$) of each index are calculated using the entropy weight method, and the calculation process is shown in Equations (1) and (2), where $X_{ij}$ is the normalized data, $i$ is the ith factor layer index, $j$ is the jth city, $n$ is the sample size, and $n = 31$.

$$e_i = -\frac{1}{lnn}\Sigma_{i=1}^{n}f_{ij}lnf_{ij}, \text{ where } f_{i_j} = \frac{X_{ij}}{\sum_{i=1}^{n}X_{ij}} \tag{1}$$

$$d_i = 1 - e_i \tag{2}$$

Subsequently, the weights of the indicators ($\omega_i$) can be calculated, as shown in Equation (3).

$$\omega_i = \frac{d_i}{\Sigma_{i=1}^{n}d_i} \tag{3}$$

Finally, the standardized values ($X_{ij}$) of each index are weighted and summed according to the index weight ($\omega_i$) to obtain the target index score, as shown in Equation (4) [56,57].

$$Y_j = \sum_{i=1, \ j=1}^{n}\omega_iX_{ij} \tag{4}$$

#### 2.4.2. Spatial Kernel Density Estimation

The spatial kernel density estimation integrate time and space factors into the kernel density estimation function, measuring the probability density of random variables and using the continuous density function to describe the morphology of random variables

under the condition of time and space synchronization, which has the advantage of a two-dimensional analysis of space and time. Its specific estimation function [58] is

$$f(x,y) = \frac{1}{Nh_xh_y} \sum_{i=1}^{N} K_x\left(\frac{X_i - x}{h_x}\right) K_y\left(\frac{Y_i - y}{h_y}\right) \qquad (5)$$

In the formula, $f(x,y)$ is the joint density function of $x$ and $y$, $N$ is the number of observed values, $h$ is the bandwidth, $K$ is the kernel function, $K_x = \frac{1}{\sqrt{2\pi}}exp\left(-\frac{x^2}{2}\right)$, $K_y = \frac{1}{\sqrt{2\pi}}exp\left(-\frac{y^2}{2}\right)$, $X_i$ and $Y_i$ are random variables, and $x$ and $y$ are the mean of the random variables.

The spatial kernel density estimation can reveal the spatiotemporal difference of urban CTDP in the research area according to three aspects. Firstly, the distribution dynamics of the CTDP in a certain city are measured from year t to year $t + 3$. Secondly, the mutual influence of the CTDP of cities in the research area under the condition of spatial lag is analyzed. Thirdly, the influence of the t year CTDP of each city in the research area on the $t + 3$ year CTDP of the city under the condition of spatial lag is analyzed.

### 2.4.3. Centroid Transferring Model

The centroid transferring model is a method to quantitatively express spatiotemporal evolution using the principle of the center of gravity. The specific calculation steps of the centroid transferring model are as follows [59]. Firstly, measure the geometric center of each polygon in the research area. Secondly, calculate the weighted centroid of the geographical attributes of each polygon according to the traditional center model, as shown in Formula (6). Thirdly, measure the transferring distance of weighted centroid of geographical attributes, as shown in Formula (7), and the time distribution equilibrium of geographical attributes in the research area can be judged according to its size. Fourthly, calculate the transferring direction of weighted centroid of geographical attributes, as shown in Formula (8), which can accurately express the differentiation of geographical attributes in different periods. Fifthly, draw the trajectory map of the transferring of geographic attributes to realize the visualization of the spatiotemporal differentiation of geographical attributes.

$$\overline{x} = \frac{1}{n} \sum_{n=1}^{n} x_i \, , \quad \overline{y} = \frac{1}{n} \sum_{n=1}^{n} y_i \qquad (6)$$

$$|L_i| = \sqrt{(x_{i+1} - x_i)^2 + (y_{i+1} - y_i)^2} \qquad (7)$$

$$\theta_i = n\frac{\pi}{2} + arctan\left(\frac{y_{i+1} - y_i}{x_{i+1} - x_i}\right) \qquad (8)$$

In the formula, $\overline{x}$ represents the horizontal coordinate of the weighted centroid of gravity of geographical attributes; $\overline{y}$ represents the vertical coordinate of the weighted centroid of gravity of geographical attributes; $x_i$ represents the horizontal coordinate of the geometric center of gravity of the $i$-th polygon; $y_i$ represents the vertical coordinate of the geometric center of gravity of the $i$-th polygon; and $n$ represents the number of polygons in the research area. $L_i$ represents the transferring distance of the weighted centroid of geographical attributes, and $\theta_i$ represents the transferring direction of the weighted centroid of geographical attributes.

In this paper, the centroid-transferring curve-based spatiotemporal pattern [59] is used to discuss the moving track of the center of the cultural tourism potential of cities in the middle and lower reaches of the Yellow River. The trajectory curve formed by the weighted centroid in different periods is measured to describe the distance and direction of centroid transferring of the CTDP in the middle and lower reaches of the Yellow River. The spatiotemporal evolution of the CTDP in the middle and lower reaches of the Yellow River is quantitatively analyzed from the two dimensions of spatial pattern and time

process. Then, the data processing and editing functions of ArcGIS are used to realize the centroid-transferring trajectory and visualization of the CTDP from 2009 to 2020.

## 3. Evaluation Model

### 3.1. Theoretical Framework

The evaluation of urban CTDP is systematic and difficult to measure using a univariate index. This paper follows the principles of scientificity, validity, comprehensiveness, compatibility, representativeness and operability of the evaluation of cultural tourism development potential according to the evaluation index system of the tourism development potential of provinces along the Belt and Road in China built by Yuying Chen [13]. Then, it refers to the relevant evaluation indexes in the existing research achievements of cultural development potential, tourism development potential and heritage tourism development potential [28,34,37–41], and four-rule layers, including the supply potential of cultural tourism, environmental support potential, cultural tourism consumption demand potential and scientific and technological innovation potential, are constructed to conclude and deduce the logical relationship among the rule layers, which can form a theoretical framework for evaluating CTDP in this study, as shown in Figure 2.

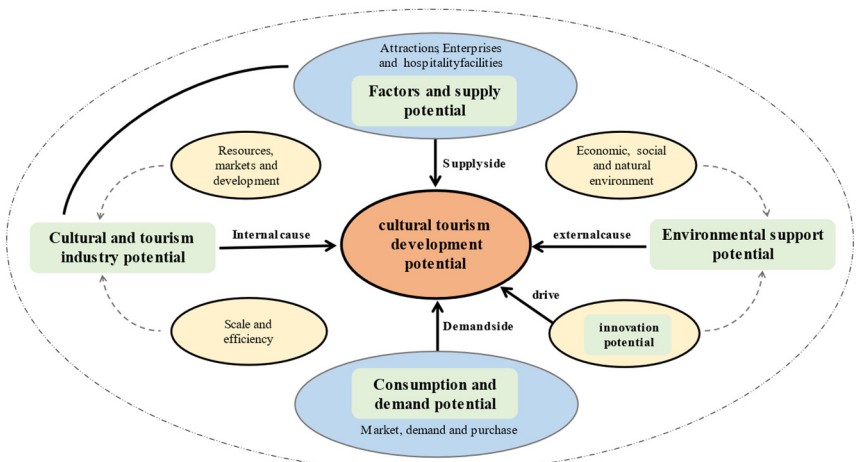

**Figure 2.** Theoretical framework of evaluation of CTDP.

### 3.1.1. Environmental Support Potential

The development of cultural tourism is relatively strongly dependent on infrastructure, economic foundations, the ecological environment and social and environmental factors, as external factors have a significant impact on sustainable CTDP [35,60]. Many scholars believe that the supply, guarantee, support and carrying capacity of the cultural tourism development environment are important indicators for development potential evaluation [61,62]. Therefore, the supply and support potential of the cultural tourism development environment is mainly manifested in the supply and support capacity of the public environment, the natural environment and the economic environment. The stronger the environmental support ability is, the better the development conditions of cultural tourism are, the stronger the appeal to consumers will be, the more developed cultural tourism will be, and the greater the sustainable development potential of regional cultural tourism will be. Among them, the public environment supply capacity reflects the regional government's infrastructure supply for the development of the cultural tourism industry, including public utilities and the basic facilities of modern social life. In this study, per capita urban road area, highway passenger volume and civil aviation passenger volume are selected to quantitatively describe the status quo and carrying capacity of regional cultural tourism traffic. The harmless treatment rate of household garbage is selected to quantify the public environmental sanitation management ability. The total volume

of telecommunication services is selected to describe the level of communication facility construction in the region.

The support capacity of the natural environment refers to the ability of the natural ecological environment to attract and safeguard cultural tourism activities. On the one hand, the seasonality of cultural tourism activities is strong, and the climate plays a certain supporting role in cultural tourism activities. The climatic characteristics can be used to describe the suitable period of cultural tourism activities [63]. On the other hand, as people pay increasing attention to the natural environment, such as air quality, vegetation and greenery, smell and noise, the natural environment has gradually become an important factor affecting the attraction and competitiveness of regional cultural tourism. The per capita proportion of regional environment and resources can represent the carrying capacity of natural environment and tourism reception levels [64].

The support capacity of the economic environment is the comprehensive supporting force of the overall regional economic development level, industrial investment and management level, service industry status and degree of openness to the outside world. In the process of regional cultural tourism development, the higher the overall level of economic development is, the more conducive it is to the development potential of regional cultural tourism. The ability of industrial investment and management to support the development of cultural tourism can be measured by indicators such as the proportion of fixed investment, the total amount of actually utilized foreign investment, the fixed asset investment in municipal public facility construction and the urbanization rate. The proportion of the tertiary industry to GDP is used to measure the economic output capacity and development level of the regional cultural tourism. The degree of openness to the outside world can be measured by the total amount of actually utilized foreign capital to reflect the urban international CTDP.

### 3.1.2. Demand Potential of Cultural Tourism

In this study, the scale of cultural tourism demand is measured by the domestic tourist arrivals and inbound tourist arrivals. Per capita GDP, Engel coefficient and price index of residents' education, culture and entertainment consumption are selected to measure the demand potential of cultural tourism. As a spiritual demand, cultural tourism can only be produced when residents' living standards reach a certain level, and regional living standards and individual economic capacity can be measured by per capita GDP [65]. The Engel coefficient and per capita expenditure on education, culture and entertainment reflect the cultural tourism expenditure as negative and positive indicators, respectively. The price index of residents' education, culture and entertainment consumption reflects the consumption capacity of regional cultural tourism and the development level of the cultural tourism economy. Per capita expenditure is the ratio between cultural tourism income and tourist number, which can measure the possible market benefits created by cultural tourism. The higher the per capita expenditure is, the better the development potential of regional cultural tourism will be.

### 3.1.3. Supply Potential of Cultural Tourism

The self-development and growth of an industry is the dominant and leading element in its development potential. The supply, consumption and output of cultural tourism enterprises are the direct driving force for the development of the cultural tourism industry. As the "internal cause" of the development potential of regional cultural tourism, the scale and benefit of this driving force are directly related to the quality of the supply of cultural tourism [66]. The scale potential of the cultural tourism industry represents the sum of material products and services that can be provided to consumers in a certain period, which is mainly affected and restricted by the number of cultural tourism enterprises and institutions. Cultural tourism enterprises can provide places for cultural tourism activities and become cultural tourism attractions. At the same time, as reception facilities, these enterprises reflect the regional cultural tourism reception ability and are the realization

condition of the development ability of the cultural tourism industry. The benefit potential of cultural tourism is the economic contribution and employment contribution of cultural tourism development to regional sustainable development. Total tourism income, tourism foreign exchange income and the operating profits of culture, sports and the entertainment industry above designated scale are the factor layers to measure the economic contribution of cultural tourism. The contribution of cultural tourism employment reflects the use of labor resources in the field of cultural tourism to a certain extent. Therefore, this study selects the proportion of employment in the tertiary industry, the number of employees in culture, sports and the entertainment industry and the number of employees in the accommodation and catering industries to measure the direct or indirect employment contribution of cultural tourism.

### 3.1.4. Technology Innovation Potential

The progress of technology makes its fields of application continue to expand. The development of cultural tourism has become one of the new fields of technology innovation. Internet technology has particularly promoted the rapid development of online cultural tourism economic activities, whose scale benefit has exceeded the physical cultural tourism economy. At present, the technological innovation ability of cultural tourism has become the key to the survival of the fittest of cultural tourism and is one of the important means to improve the market competitiveness. The sustainable development of regional cultural tourism depends on scientific progress and technological innovation. The human resources, capital investment and patent rights of cultural tourism research and development can accurately reflect the scientific and technological innovation ability of cultural tourism [67]. Therefore, the scientific and technological innovation potential of regional cultural tourism can be evaluated using three indicators: the number of R&D personnel, the R&D expenditure and the number of patent grants.

### *3.2. Index System*

The common alternative approaches to development potential evaluation include SWOT, descriptive analysis, item response theory, GIS, item response theory model, and stakeholders' assessment. Most of these methods perform their evaluation from the perspective of supply. The dominant role of supply evaluation factors leads to the one-sidedness of the evaluation results [13]. In order to avoid these problems, some studies add a corresponding conceptual factor layer of the research object into the development potential model, but this makes it difficult for relevant research to be evaluated from a purely market-based perspective [68]. In this study, the entropy weight method is used to obtain the entropy and difference coefficient of each factor and to determine the weight of the factor, as shown in Table 1.

Among the four-rule layers in Table 1, the technological innovation potential of cultural tourism is the most important to the development potential of municipal cultural tourism, with a weight of 34.86%, and is the primary driving force for cultural tourism in the middle and lower reaches of the Yellow River. The second is the supply potential of cultural tourism, with a weight of 24.57%; the environmental support potential and demand potential of cultural tourism are of basically the same importance. Among the 31 factor layers in Table 1, the most important one is civil aviation passenger volume (8.57%), followed by the number of cultural tourism patent rights (5.77%). The TOP 10 are civil aviation passenger volume, patent rights of cultural tourism, foreign revenue of tourism, construction of municipal public facilities, number of 5A scenic spots, total amount of actually utilized foreign capital, total amount of telecommunication businesses, inbound tourist number, R&D expenditure, and number of R&D personnel. There are 12 factor layers with an importance higher than 3.5%.

**Table 1.** Three-level index system of evaluation of CTDP.

| Rule Layers | Sub-Rule Layers | Factor Layers | Weight |
|---|---|---|---|
| Environmental support potential ($X_1$) 20.95% | Public environment supply capacity ($X_{11}$) 15.25% | Highway passenger volume ($X_{111}$) | 2.90% |
| | | Civil aviation passenger volume ($X_{112}$) | 8.57% |
| | | Per capita urban road area ($X_{113}$) | 2.13% |
| | | Harmless treatment rate of household garbage ($X_{114}$) | 1.48% |
| | | Total telecommunication services ($X_{115}$) | 4.03% |
| | Natural environment support capacity ($X_{12}$) 6.43% | Days with good air quality ($X_{121}$) | 1.82% |
| | | Green coverage in built-up areas ($X_{122}$) | 1.48% |
| | | Park land per capita ($X_{123}$) | 1.58% |
| | | Regional population density ($X_{124}$) | 1.56% |
| | Economic environment support capacity ($X_{13}$) 12.10% | Proportion of the tertiary industry in the gross regional product ($X_{131}$) | 1.80% |
| | | Proportion of fixed investment in the tertiary industry ($X_{132}$) | 1.68% |
| | | Total amount of actually utilized foreign investment ($X_{133}$) | 4.89% |
| | | Fixed asset investment in municipal public facility construction ($X_{134}$) | 5.06% |
| | | Urbanization rate ($X_{135}$) | 1.71% |
| Demand potential of cultural tourism ($X_2$) 19.62% | Cultural tourism demand capacity ($X_{21}$) 14.57% | Domestic tourist arrivals ($X_{211}$) | 2.59% |
| | | Inbound tourist arrivals ($X_{212}$) | 4.71% |
| | Cultural tourism purchasing power ($X_{22}$) 6.52% | Engel coefficient ($X_{221}$) | −1.49% |
| | | Per capita GDP ($X_{222}$) | 2.00% |
| | | Per capita expenditure on education, culture and entertainment ($X_{223}$) | 1.41% |
| Supply potential of cultural tourism ($X_3$) 24.57% | Industry scale contribution ($X_{31}$) 12.78% | Number of high A-level scenic spots ($X_{311}$) | 4.91% |
| | | Number of enterprises in the accommodation and catering industries ($X_{312}$) | 3.57% |
| | | Number of public toilets ($X_{313}$) | 2.08% |
| | | Number of cultural heritage sites ($X_{314}$) | 2.51% |
| | | Number of cultural institutions ($X_{315}$) | 2.93% |
| | Economic benefit potential ($X_{32}$) 13.62% | Total tourism revenue ($X_{321}$) | 3.05% |
| | | Tourism foreign exchange revenue ($X_{322}$) | 5.57% |
| | | Proportion of employment in the tertiary industry ($X_{323}$) | 1.79% |
| | | Number of employees in culture, sports and the entertainment industry ($X_{324}$) | 2.94% |
| | | Number of employees in the accommodation and catering industries ($X_{325}$) | 3.70% |
| Technological innovation potential ($X_4$) 34.86% | Technological innovation potential ($X_{41}$) 18.73% | Number of R&D personnel ($X_{411}$) | 3.90% |
| | | R&D expenditure ($X_{412}$) | 4.40% |
| | | Number of patent grants ($X_{413}$) | 5.77% |

Data source: collated and calculated by the author.

### 3.3. CTDP Index Model of the Middle and Lower Reaches of the Yellow River

This study uses a multi-index evaluation model to measure the CTDP of cities in the middle and lower reaches of the Yellow River. The basic principle of this evaluation model is shown in Equation (4). According to this principle, this paper firstly retrieves and processes the index data of each factor layer of 43 cities in the case area from 2009 to 2020, and then uses the entropy method to calculate the weights and scores step by step. The target layer scores can be obtained according to the weights and scores of the main criterion layer, that is, the comprehensive index of the cultural tourism development potential of each city, as shown in Equation (9).

$$Y = 0.2095X_1 + 0.1962X_2 + 0.2475X_3 + 0.3486X_4 \tag{9}$$

$Y$ is the CTDP, and a higher value of $Y$ indicates greater development potential. $\omega_i$ is the weight of the i-th rule layers. As shown in Table 1, $\omega_1 = 0.2095$, $\omega_2 = 0.1962$, $\omega_3 = 0.2475$ and $\omega_4 = 0.3486$. $X_i$ is the comprehensive score of the i-th development potential evaluation rule layers. Among them, $X_1$ is the environmental support potential, $X_2$ is the demand potential of cultural tourism, $X_3$ is the supply potential of cultural tourism, and $X_4$ is the technological innovation potential.

## 4. Results

### 4.1. The CTDP Scores of Cities in the Middle and Lower Reaches of the Yellow River Continue to Improve

According to Formula (6), the CTDP of 43 cities in the middle and lower reaches of the Yellow River is calculated, as shown in Table 2. A higher score of the CTDP indicates a higher development potential. The CTDP of all cities shows an upward trend from 2009 to 2019 and reaches a peak around 2019, with a maximum value of 0.6709.

**Table 2.** CTDP score of cities in the middle and lower reaches of the Yellow River.

| City | 2009 | 2010 | 2011 | 2012 | 2013 | 2014 |
|------|------|------|------|------|------|------|
| Taiyuan | 0.1907 | 0.2003 | 0.2153 | 0.2290 | 0.2520 | 0.2453 |
| Datong | 0.1282 | 0.1306 | 0.1320 | 0.1420 | 0.1488 | 0.1530 |
| Yangquan | 0.1041 | 0.1172 | 0.1184 | 0.1225 | 0.1215 | 0.1225 |
| Changzhi | 0.1121 | 0.1248 | 0.1209 | 0.1363 | 0.1469 | 0.1597 |
| Jincheng | 0.1227 | 0.1330 | 0.1433 | 0.1506 | 0.1597 | 0.1665 |
| Shuozhou | 0.1153 | 0.1181 | 0.1251 | 0.1284 | 0.1393 | 0.1412 |
| Jinzhong | 0.1016 | 0.1080 | 0.1136 | 0.1237 | 0.1486 | 0.1712 |
| Yuncheng | 0.1089 | 0.1196 | 0.1245 | 0.1316 | 0.1535 | 0.1553 |
| Xinzhou | 0.1091 | 0.1121 | 0.1091 | 0.1212 | 0.1343 | 0.1440 |
| Linfen | 0.1012 | 0.1224 | 0.1228 | 0.1116 | 0.1377 | 0.1492 |
| Lvliang | 0.1429 | 0.1315 | 0.1372 | 0.1415 | 0.1495 | 0.1576 |
| Tianshui | 0.1216 | 0.1240 | 0.1360 | 0.1388 | 0.1327 | 0.1370 |
| Qingyang | 0.1183 | 0.1134 | 0.1188 | 0.1257 | 0.1298 | 0.1240 |
| Pingliang | 0.1265 | 0.1203 | 0.1267 | 0.1319 | 0.1380 | 0.1392 |
| Xi'an | 0.3046 | 0.3470 | 0.3957 | 0.4066 | 0.4281 | 0.4535 |
| Tongchuan | 0.1183 | 0.1169 | 0.1096 | 0.1205 | 0.1200 | 0.1173 |
| Baoji | 0.1226 | 0.1284 | 0.1373 | 0.1393 | 0.1477 | 0.1617 |
| Xianyang | 0.1045 | 0.1179 | 0.1269 | 0.1366 | 0.1487 | 0.1438 |
| Weinan | 0.1249 | 0.1269 | 0.1448 | 0.1463 | 0.1551 | 0.1525 |
| Yan'an | 0.1187 | 0.1220 | 0.1277 | 0.1302 | 0.1411 | 0.1370 |
| Yulin | 0.1099 | 0.1201 | 0.1246 | 0.1339 | 0.1410 | 0.1547 |
| Shangluo | 0.1143 | 0.1248 | 0.1362 | 0.1287 | 0.1186 | 0.1231 |
| Luoyang | 0.1587 | 0.1653 | 0.2034 | 0.2370 | 0.2613 | 0.2668 |
| Jiaozuo | 0.1228 | 0.1462 | 0.1489 | 0.1547 | 0.1622 | 0.1637 |
| Sanmenxia | 0.1313 | 0.1165 | 0.1247 | 0.1262 | 0.1278 | 0.1317 |
| Zhengzhou | 0.2520 | 0.2603 | 0.2774 | 0.2891 | 0.3162 | 0.3089 |
| Kaifeng | 0.1256 | 0.1242 | 0.1396 | 0.1508 | 0.1414 | 0.1495 |
| Anyang | 0.1257 | 0.1276 | 0.1400 | 0.1403 | 0.1470 | 0.1382 |
| Hebi | 0.1116 | 0.1182 | 0.1183 | 0.1215 | 0.1278 | 0.1211 |
| Xinxiang | 0.1202 | 0.1232 | 0.1282 | 0.1363 | 0.1390 | 0.1395 |
| Puyang | 0.0975 | 0.0914 | 0.0926 | 0.0892 | 0.1029 | 0.1039 |
| Shangqiu | 0.1056 | 0.1045 | 0.1085 | 0.1129 | 0.1212 | 0.1199 |
| Jinan | 0.2083 | 0.2138 | 0.2351 | 0.2464 | 0.2674 | 0.2641 |
| Qingdao | 0.2461 | 0.2751 | 0.3019 | 0.3378 | 0.3472 | 0.3622 |
| Zibo | 0.1709 | 0.1803 | 0.1903 | 0.1969 | 0.1929 | 0.1690 |
| Dongying | 0.1172 | 0.1194 | 0.1245 | 0.1374 | 0.1470 | 0.1515 |
| Weifang | 0.1556 | 0.1665 | 0.1695 | 0.1779 | 0.2533 | 0.1887 |
| Jining | 0.1441 | 0.1722 | 0.1595 | 0.1698 | 0.1762 | 0.1729 |
| Tai'an | 0.1301 | 0.1458 | 0.1562 | 0.1640 | 0.1727 | 0.1763 |

**Table 2.** *Cont.*

| City | 2009 | 2010 | 2011 | 2012 | 2013 | 2014 |
|------|------|------|------|------|------|------|
| Dezhou | 0.1233 | 0.1220 | 0.1157 | 0.1386 | 0.1486 | 0.1514 |
| Liaocheng | 0.1113 | 0.1088 | 0.1098 | 0.1193 | 0.1294 | 0.1306 |
| Binzhou | 0.0957 | 0.1087 | 0.1206 | 0.1280 | 0.1406 | 0.1437 |
| Heze | 0.1039 | 0.1041 | 0.1153 | 0.1189 | 0.1183 | 0.1158 |
| Taiyuan | 0.1907 | 0.2003 | 0.2153 | 0.2290 | 0.2520 | 0.2453 |
| Datong | 0.1282 | 0.1306 | 0.1320 | 0.1420 | 0.1488 | 0.1530 |
| Yangquan | 0.1041 | 0.1172 | 0.1184 | 0.1225 | 0.1215 | 0.1225 |
| Changzhi | 0.1121 | 0.1248 | 0.1209 | 0.1363 | 0.1469 | 0.1597 |
| Jincheng | 0.1227 | 0.1330 | 0.1433 | 0.1506 | 0.1597 | 0.1665 |
| Shuozhou | 0.1153 | 0.1181 | 0.1251 | 0.1284 | 0.1393 | 0.1412 |
| Jinzhong | 0.1016 | 0.1080 | 0.1136 | 0.1237 | 0.1486 | 0.1712 |
| Yuncheng | 0.1089 | 0.1196 | 0.1245 | 0.1316 | 0.1535 | 0.1553 |
| Xinzhou | 0.1091 | 0.1121 | 0.1091 | 0.1212 | 0.1343 | 0.1440 |
| Linfen | 0.1012 | 0.1224 | 0.1228 | 0.1116 | 0.1377 | 0.1492 |
| Lvliang | 0.1429 | 0.1315 | 0.1372 | 0.1415 | 0.1495 | 0.1576 |
| Tianshui | 0.1216 | 0.1240 | 0.1360 | 0.1388 | 0.1327 | 0.1370 |
| Qingyang | 0.1183 | 0.1134 | 0.1188 | 0.1257 | 0.1298 | 0.1240 |
| Pingliang | 0.1265 | 0.1203 | 0.1267 | 0.1319 | 0.1380 | 0.1392 |
| Xi'an | 0.3046 | 0.3470 | 0.3957 | 0.4066 | 0.4281 | 0.4535 |
| Tongchuan | 0.1183 | 0.1169 | 0.1096 | 0.1205 | 0.1200 | 0.1173 |
| Baoji | 0.1226 | 0.1284 | 0.1373 | 0.1393 | 0.1477 | 0.1617 |
| Xianyang | 0.1045 | 0.1179 | 0.1269 | 0.1366 | 0.1487 | 0.1438 |
| Weinan | 0.1249 | 0.1269 | 0.1448 | 0.1463 | 0.1551 | 0.1525 |
| Yan'an | 0.1187 | 0.1220 | 0.1277 | 0.1302 | 0.1411 | 0.1370 |
| Yulin | 0.1099 | 0.1201 | 0.1246 | 0.1339 | 0.1410 | 0.1547 |
| Shangluo | 0.1143 | 0.1248 | 0.1362 | 0.1287 | 0.1186 | 0.1231 |
| Luoyang | 0.1587 | 0.1653 | 0.2034 | 0.2370 | 0.2613 | 0.2668 |
| Jiaozuo | 0.1228 | 0.1462 | 0.1489 | 0.1547 | 0.1622 | 0.1637 |
| Sanmenxia | 0.1313 | 0.1165 | 0.1247 | 0.1262 | 0.1278 | 0.1317 |
| Zhengzhou | 0.2520 | 0.2603 | 0.2774 | 0.2891 | 0.3162 | 0.3089 |
| Kaifeng | 0.1256 | 0.1242 | 0.1396 | 0.1508 | 0.1414 | 0.1495 |
| Anyang | 0.1257 | 0.1276 | 0.1400 | 0.1403 | 0.1470 | 0.1382 |
| Hebi | 0.1116 | 0.1182 | 0.1183 | 0.1215 | 0.1278 | 0.1211 |
| Xinxiang | 0.1202 | 0.1232 | 0.1282 | 0.1363 | 0.1390 | 0.1395 |
| Puyang | 0.0975 | 0.0914 | 0.0926 | 0.0892 | 0.1029 | 0.1039 |
| Shangqiu | 0.1056 | 0.1045 | 0.1085 | 0.1129 | 0.1212 | 0.1199 |
| Jinan | 0.2083 | 0.2138 | 0.2351 | 0.2464 | 0.2674 | 0.2641 |
| Qingdao | 0.2461 | 0.2751 | 0.3019 | 0.3378 | 0.3472 | 0.3622 |
| Zibo | 0.1709 | 0.1803 | 0.1903 | 0.1969 | 0.1929 | 0.1690 |
| Dongying | 0.1172 | 0.1194 | 0.1245 | 0.1374 | 0.1470 | 0.1515 |
| Weifang | 0.1556 | 0.1665 | 0.1695 | 0.1779 | 0.2533 | 0.1887 |
| Jining | 0.1441 | 0.1722 | 0.1595 | 0.1698 | 0.1762 | 0.1729 |
| Tai'an | 0.1301 | 0.1458 | 0.1562 | 0.1640 | 0.1727 | 0.1763 |
| Dezhou | 0.1233 | 0.1220 | 0.1157 | 0.1386 | 0.1486 | 0.1514 |
| Liaocheng | 0.1113 | 0.1088 | 0.1098 | 0.1193 | 0.1294 | 0.1306 |
| Binzhou | 0.0957 | 0.1087 | 0.1206 | 0.1280 | 0.1406 | 0.1437 |
| Heze | 0.1039 | 0.1041 | 0.1153 | 0.1189 | 0.1183 | 0.1158 |

Data source: calculated by the author.

From the CTDP of each city, Xi'an was the city with the highest regional CTDP from 2009 to 2020, and the cities closest to the 12-year annual average value are mainly distributed in Shandong Province, such as Tai'an City in 2010, 2013 and 2018, and Zibo City in 2014–2016. In other years, it was concentrated in Baoji, Weinan, Sanmenxia, Jiaozuo, Jinzhong and other cities near the border area between Henan, Shaanxi and Shanxi. From the analysis of the fluctuation of CTDP, in the direction of time series of each city, the overall fluctuation is significant, with the smallest fluctuation range in Zibo City and the largest in Xi'an City. However, the CTDP gap between different cities is increasing year

by year. According to the average score of the urban CTDP from 2009 to 2020, Xi'an has the greatest potential, while Puyang has the smallest potential. Therefore, the urban CTDP in the middle and lower reaches of the Yellow River during 2009–2020 shows an increasing trend but is small on the whole. By analyzing the growth rate, Xi'an, Qingdao, Zhengzhou, Luoyang and Jinan are found to have the fastest potential growth; the CTDP of Jinzhong, Taiyuan, Linfen and Yuncheng in Shanxi Province has increased rapidly. However, Tianshui, Qingyang and Pingliang in Gansu Province have low CTDP and a slow growth rate. From the perspective of a horizontal comparison between cities, the profound cultural background and well-known tourism brands of Xi'an show a leading position in long-term, with its potential reaching 0.67 in 2019, while Zhengzhou and Qingdao, as the new first-tier cities, have the second-tier cultural tourism potential.

*4.2. CTDP Potential Level of the Cities in the Study Area Is Divided into Five Levels*

According to Table 2, this paper adopts the k-means clustering method for cluster analysis with the help of SPSS software 24.0. In general, the value of K is 3–8 [33,34,65–67,69]. Due to the large number of objects studied in this paper, this paper sets K as five and then adopts an iterative method according to the minimum Euclidean distance from the class center to better explore the hierarchical spatial structure characteristics. By measuring its clustering center, the CTDP score of 43 cities can be clustered into five levels [65,66,70,71], namely, a level I potential area, whose CTDP score is greater than or equal to 0.41, is a high potential area; a level II potential area, with a CTDP score ranging from 0.31 to 0.41, is a medium-high potential area; a level III potential area, with a CTDP score between 0.18 and 0.31, is a medium potential area; a level IV potential area, whose CTDP score ranges from 0.13 to 0.18, is a medium-low potential area; a level V potential area, with a CTDP score ≤ 0.13, is low potential area. A level I potential area is represented by Xi'an, Zhengzhou and Qingdao. This potential area includes new first-tier provincial capitals or sub-provincial cities with rich cultural tourism resources, developed economies, a mature cultural tourism industry and obvious geographical and transportation advantages. These cities are cultural tourism destinations with international influence and play a leading role in the regional development of the cultural tourism industry. Level II potential areas are represented by Luoyang, Taiyuan and Jinan, which are among the second-tier cities in China. They have the advantages of provincial capital policies and strategies and cultural tourism brands. However, the lack of technological innovation ability of the cities has certain constraints on the development potential of cultural tourism. A level III potential area is radiated and driven by the level I and level II potential areas, and its market demand and purchasing power are at the medium development level. It has certain industrial development foundations and cultural tourism resource endowments, but the supply of cultural tourism development and the support force of scientific and technological innovation are not enough, so it includes the cities in the transition stage to the level II potential of cultural tourism development. The economic development of level IV potential areas is uneven, but the development of cultural tourism resources is insufficient, and the cultural tourism industry cannot occupy an important position in the development strategy. It is the medium-low potential area. The economic development of level V potential areas is relatively backward, with its urban development mainly dominated by primary and secondary industries, and the development of the cultural tourism industry is not mature. Most of these areas are in the early stage of development of a level IV potential area. From level I to level V, the levels of potential areas decreased successively. The cluster analysis results of cultural tourism development potential of 43 cities in the middle and lower reaches of the Yellow River are shown in Figure 3.

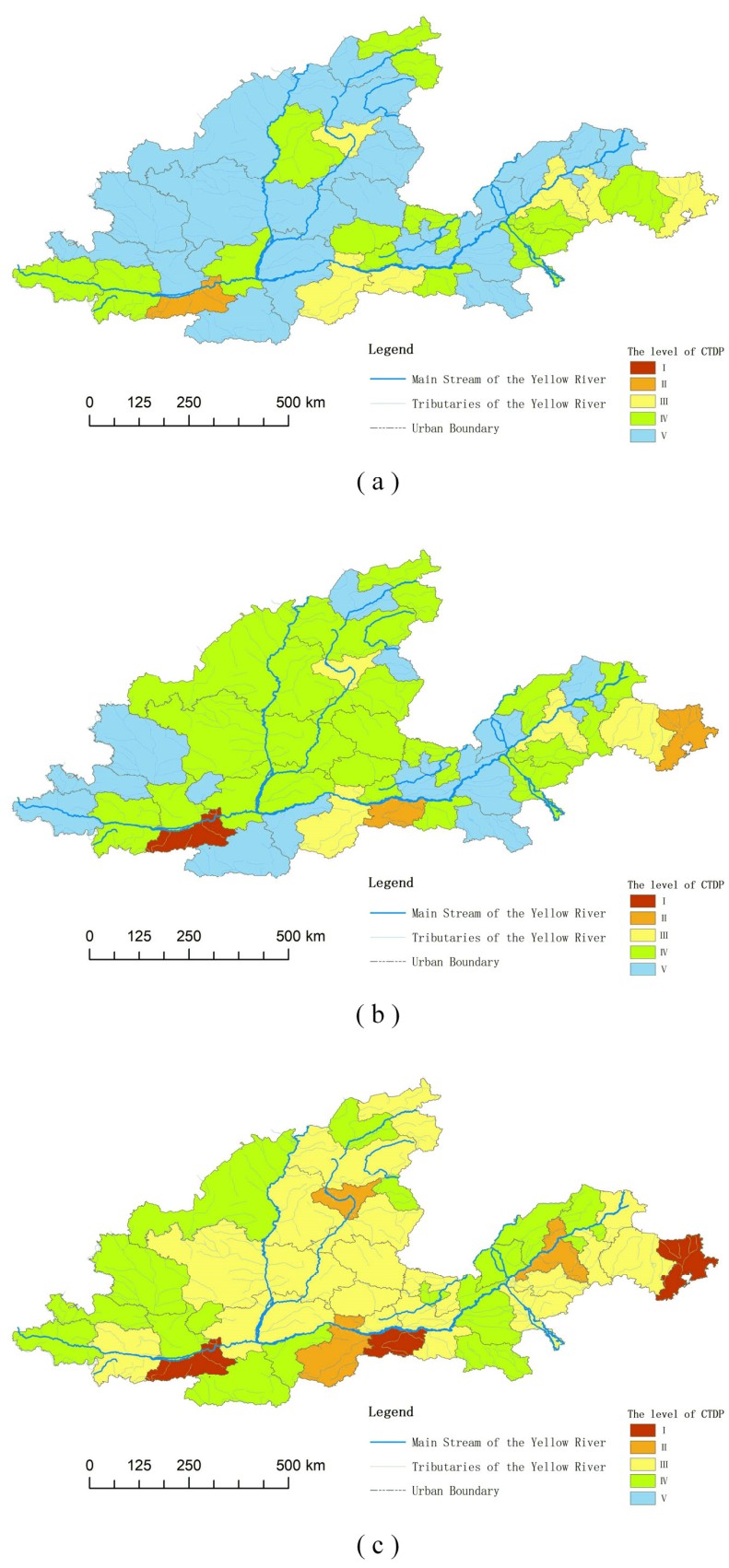

**Figure 3.** The level of CTDP of cities in the research area: (**a**) potential level from 2009 to 2012; (**b**) potential level from 2013 to 2016; (**c**) potential level from 2017 to 2020.

After a comprehensive analysis of Figure 3 and Table 2, it can be found that the evolution characteristics of urban cultural tourism development potential levels in the middle and lower reaches of the Yellow River are as follows: from 2009 to 2012, there was no level I high potential area, and the level I potential area increased from Xi'an from 2013 to 2016 to Xi'an, Zhengzhou and Qingdao from 2017 to 2020. As the main representative of Guanzhong culture, the CTDP of Xi'an always took the leading position in the three periods. The average annual potential scores of Xi'an are 0.3635, 0.4667 and 0.6089, respectively. In 2017–2020, the average annual potential values of Zhengzhou and Qingdao are 0.4213 and 0.4472, respectively, which are much lower than Xi'an. The number of level II medium-high potential areas increased from one city (Xi'an) in 2009–2012 to three cities (Jinan, Taiyuan and Luoyang) in 2017–2020. The sequence evolution of high potential areas is as follows: Xi'an increased from a level II potential area in the first period to a level I potential area in the second and third periods. Zhengzhou and Qingdao increased from level II potential areas in the second period to level I potential areas in the third period. Luoyang, Taiyuan and Jinan all changed from level III potential areas in the first and second periods to level II potential areas in the third period. The number of level III potential areas accounted for 13.95% in the first period, 9.3% in the second period, and 46.51% in the third period. The number of level V potential areas decreased from 24 in the first period to 16 in the second period and 0 in the third period. This indicates that, on the whole, the development potential of urban cultural tourism in the middle and lower reaches of the Yellow River has seen a significant increase. Relatively speaking, the growth of level III potential areas has a large fluctuation, while the growth of level V potential areas directly decreases from 55.81% to 0%, with the largest fluctuation range. However, the potential areas of level I and level II have small fluctuations, accounting for only 13.95% of the total in 2020. This indicates that the CTDP in the research area needs to be improved in order to better implement the national strategy of inheriting and promoting Yellow River culture.

### 4.3. Spatiotemporal Differentiation of CTDP in the Research Area Is Significant

If the CTDP score in Table 2 is taken as the spatial geographic attribute, the spatiotemporal differentiation of CTDP in the research area can be explored by using the spatial kernel density estimation. In order to analyze the spatiotemporal differentiation of CTDP in the research area, this paper estimates the unconditional kernel density, static kernel density and dynamic kernel density, respectively.

4.3.1. Spatial Differentiation of Potential Intensity of CTDP Based on Mean Value

According to Table 2, the arithmetic mean value of the CTDP index of each city from 2009 to 2020 is calculated. Taking it as a spatial geographic attribute, the spatial distribution of the CTDP intensity in the research area can be obtained using Kriging interpolation analysis (see Figure 4). The cities with higher average CTDP levels have higher potentials. From the overall regional analysis, there are significant spatial differences in the development potential intensity of CTDP in the middle and lower reaches of the Yellow River. The level I and level II potential areas form the potential growth pole of CTDP, and the level IV and level V potential areas interweave with each other in the whole region. From the analysis of regional differences, the potential area of cultural tourism development in Guanzhong urban agglomeration with Xi'an as the pole-center forms the spatial structure of the polar core (Xi'an)—trough (Baoji–Xianyang–Weinan), which is in the core area of the middle and lower reaches of the Yellow River. The Central Plains urban agglomeration with Zhengzhou as the core and the Jinan–Qingdao metropolitan area are the secondary centers; the Taiyuan city agglomeration with Taiyuan as the center shows a significant trend of agglomeration; the development potential of urban cultural tourism in the Ganshan–Shanxi border and Yu–Lu border areas is small, far from the high-potential centers, and the degree of dispersion is high.

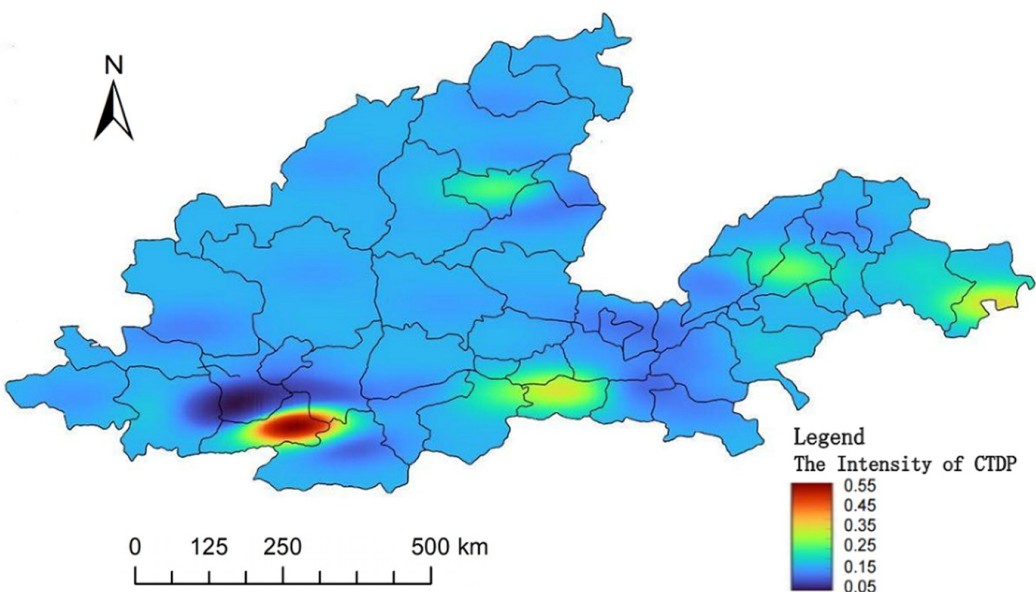

**Figure 4.** Spatial differentiation of development potential intensity of cultural tourism.

### 4.3.2. Spatiotemporal Differentiation of CTDP Based on Time Series

According to Formula (1), the kernel density score of the CTDP of each city is calculated by means of ArcGIS10.6, and then the spatial kernel density of the middle and lower reaches of the Yellow River is obtained through Kriging interpolation, as shown in Figure 5.

In Figure 5a, the CTDP score in the t year is $x$, and CTDP score in the $t + 3$ year is $y$, which are substituted into Formula (1) to obtain the influence of time on urban CTDP. The greater the kernel density value is, the greater the change in urban CTDP is over time. In Figure 5b, the CTDP score of a target city is $x$, and the CTDP score of other cities in the same year is $y$, and they are substituted into Formula (1) to obtain the mutual influence of the CTDP between cities. The greater the kernel density value is, the greater the CTDP score is affected by the interaction between cities. In Figure 5c, the CTDP score of a target city in $t + 3$ years is $x$, and the CTDP score of other cities in $t + 3$ years is $y$, which are substituted into Formula (1) to obtain the dynamic influence of the CTDP of other cities on the CTDP of the target city. The greater the kernel density score is, the greater the influence of other cities on the CTDP in the target city is over time.

From the analysis of time's degree of influence on the urban CTDP (as shown in Figure 5a), it can be seen that the CTDP of Xi'an and Qingdao changed little over time and is relatively stable, while the CTDP of the central and eastern cities of Henan, the inland cities of eastern Shandong and the cities in northern Shanxi and central Shanxi all fluctuate to a certain extent. From the analysis of time series, the development potential fluctuates most in the cities of northwest Shanxi, northern Henan and eastern Shandong. From the mutual influence of the CTDP among cities (as shown in Figure 5b), it can be seen that Xi'an has the weakest correlation with the surrounding cities, followed by Qingdao, Zhengzhou and Taiyuan, while Jinan, Datong, Luliang and Yulin have relatively significant correlation with the surrounding cities, indicating that the development of cultural tourism in these cities has a strong regional dependence. From the degree of influence by the CTDP change in other cities (as shown in Figure 5c), Xi'an and Qingdao are least affected by the change in time series of the CTDP in other cities, followed by Luoyang. Zhengzhou, Jinan and Taiyuan are more affected by the CTDP change in other cities to a certain extent. The cities in western Shandong, northern Henan, northwest Shanxi and Gansu are most affected by the changes in other cities.

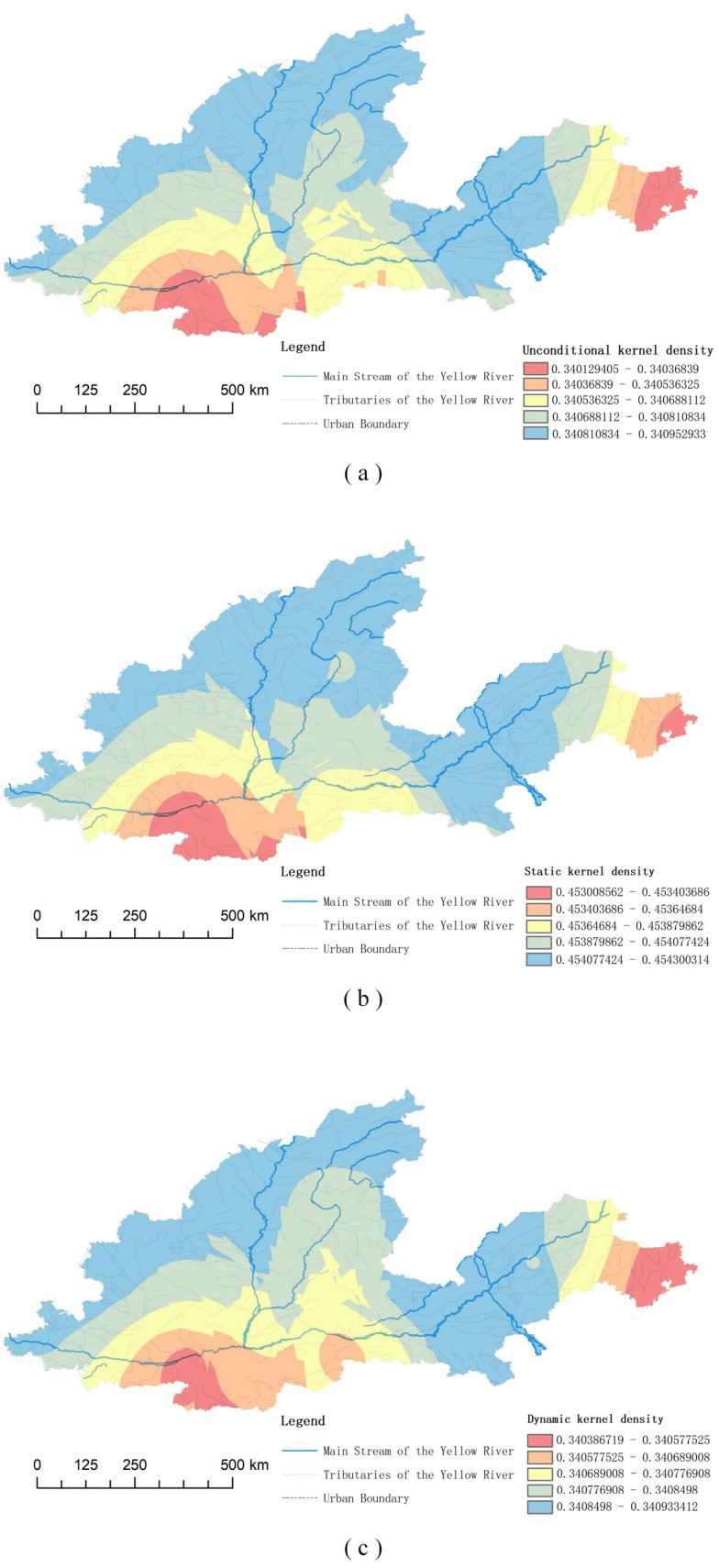

**Figure 5.** Kernel density of CTDP in the research area: (**a**) unconditional kernel density; (**b**) static kernel density; (**c**) dynamic kernel density.

### 4.4. Potential Centroid of CTDP in the Research Area Is Gradually Moving toward the Geometric Center

According to the geometric center coordinates of 43 cities in the middle and lower reaches of the Yellow River, firstly, the traditional center model (see Formula (2)) is used to calculate the weighted centroid of CTDP of each city. Secondly, according to Formula (3), the transferring distance of the weighted centroid of the CTDP is measured, and the equilibrium degree of the time distribution of the CTDP in the middle and lower reaches of the Yellow River is judged according to its size. Thirdly, according to Formula (4), the transferring direction of weighted centroid of CTDP is calculated, to express scientifically and accurately the differentiation of CTDP in different periods. Fourthly, the track chart of the transferring of the centroid of CTDP is drawn. The measured centroid of CTDP and its transferring trajectory are shown in Figure 6, and the measured results of transferring distance and direction are shown in Table 3.

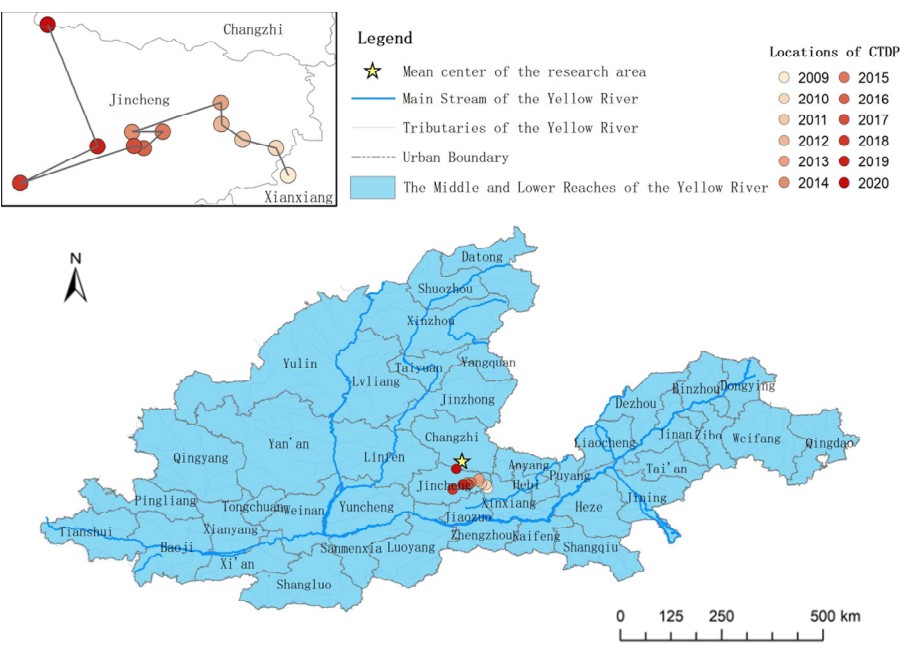

**Figure 6.** Location and transferring trajectory of CTDP in the research area.

**Table 3.** Measured scores of transferring distance and direction of CTDP in the research area.

| Year | Transferring Distance (m) | Transferring Angle | Bearing Angle Change | Year | Transferring Distance (m) | Transferring Angle | Bearing Angle Change |
|------|---------------------------|--------------------|----------------------|------|---------------------------|--------------------|----------------------|
| 2009–2010 | 8415.72 | - | Northwest 47.62 | 2015–2016 | 6481.68 | 18.90 | Northwest 08.95 |
| 2010–2011 | 8255.83 | 143.30 | Northwest 84.28 | 2016–2017 | 2410.72 | 156.29 | Northwest 85.26 |
| 2011–2012 | 6597.72 | 170.02 | Northwest 74.28 | 2017–2018 | 28,850.90 | 168.18 | Northwest 97.08 |
| 2012–2013 | 6334.98 | 109.68 | Northwest 3.92 | 2018–2019 | 21,081.17 | 3.30 | Northwest 79.58 |
| 2013–2014 | 22,551.09 | 86.51 | Northwest 97.34 | 2019–2020 | 36,858.99 | 53.72 | Northwest 46.66 |
| 2014–2015 | 7201.41 | 7.33 | Northwest 89.96 | - | - | - | - |

Data source: calculated by the author.

It can be seen from Table 3 and Figure 6 that the potential centroid of CTDP in the research area is gradually moving toward the geometric center, and its spatial moving track features are as follows. (1) The spatiotemporal pattern of potential centroid transferring shows the characteristics of multi-directional movement along the line of northwest–southwest–northeast–due north, and it gathers in the Xinxiang–Jincheng–Changzhi urban circle with Jincheng as the core. The reasons are that, although the neighboring cities of Henan, Shaanxi and Shanxi provinces have a rich cultural background, with a high density and high quality of tourism resources from the perspective of environmental support potential and factor supply potential of CTDP, the downstream Shandong Province has been

continuously superior since the Ninth Five-Year Plan of China's national economy, while since the Twelfth Five-Year Plan period, the environmental support potential and factor supply potential of Henan, Shaanxi and Shanxi provinces have increased significantly. In particular, cultural tourism in Zhengzhou, Xi'an, Taiyuan, Luoyang, Jiaozuo, Jinzhong, Weinan and other cities has developed rapidly. At the same time, the market law of cultural tourism demand is growing rapidly across the country, the scale of the tourist market of the countries along the Belt and Road who prefer Chinese culture has increased significantly, and the cities in the research area with richer historical and cultural resources are more attractive. The high-speed railway line with Zhengzhou as the hub of communication has better transport conditions than cities in downstream Shandong Province. (2) The transferring trajectory of the CTDP clustering period of each city is significantly different. From 2009 to 2012, the spatial distance of the centroid transferring is small and relatively concentrated, and it moves significantly to the northwest direction. From 2013 to 2016, there was a big difference in the spatial distance of the centroid transferring. From 2013 to 2014, the spatial distance of the centroid transferring was large, but from 2014 to 2016, the centroid transferred a very small distance, during which stage the CTDP in the research area transferred significantly to the southwest. From 2017 to 2020, the centroid transferring covered a large spatial distance, and the direction of movement was more scattered, showing a multi-directional movement track. (3) There are significant differences in the distance and angle of the centroid transferring of CTDP in the research area over the years, which indicates that there are differences in the cooperation and collaborative development of cultural tourism development among cities. (4) The spatial aggregation area of the potential centroid in the research area over the 12 years from 2009 to 2020 is located in the Zhongyuan urban agglomeration.

## 5. Conclusions and Discussions

### 5.1. Conclusions

Based on the principle of comprehensive linear function evaluation, an evaluation model of the CTDP of the research area is constructed, namely, $Y = 0.2095X_1 + 0.1962X_2 + 0.2475X_3 + 0.3486X_4$. The evaluation model of the CTDP is only applicable to the study area, but the principle, method and process of its model construction are universal and have reference significance for the study of the development potential of regional cultural tourism.

The CTDP shows an overall increasing trend over time. Over the 12 years from 2009 to 2020, the score of urban CTDP in the research area showed an upward trend, and the development potential continued to be enhanced. Each city's growth rate score is different, and most cities have slight fluctuations in the upward trend, which is consistent with the characteristics of the sustainable tourism development potential of Dunhuang City in the upper reaches of the Yellow River [64]. Due to social and economic development and scientific and technological progress in the Yellow River Basin, the public environmental supply of cities has become complete, the quality of the natural environment has been continuously improved, the economic support capacity has gradually increased, the demand and purchasing power of tourists has increased, and the supply of cultural tourism industry has met the demand. Its scale and benefits have been continuously improved. The society has paid more and more attention to scientific and technological innovation, and the development of cultural tourism has become more intelligent.

The CTDP scores of 43 cities in the research area can be clustered into five different levels (from level I to level V). Level I is a high-potential area with a high density and high quality of cultural tourism resources, a high level of economic development, a long industrial chain, a good investment environment, strong scientific and technological innovation ability and obvious market location and transportation advantages. It is generally a destination with a certain international influence and has a strong radiation driving effect. Level II is a medium-high potential area, which has a large density and high quality of cultural tourism resources, a good economic development foundation, a reasonable

industrial structure, a good investment environment, a certain market competitiveness and the strategic advantages of regional central city development and cultural tourism brand, but the urban scientific and technological innovation ability is insufficient, with significant constraints on the CTDP. Level III is a medium potential area, endowed with a certain industrial development foundation and cultural tourism resources, but it is difficult to develop, with a location at the edge of the high-end tourist market, little policy support from cultural tourism development, and a lack of scientific and technological innovation and other supporting forces, though level III has the development strength to transition to a level II potential area. Level IV is medium low potential area, which has insufficient development of cultural tourism resources, a low level of economic development, underdeveloped commercial trade, a low national economic status in the cultural tourism industry, little policy support and a weak capacity for scientific and technological innovation. Level V is a low potential area, which is relatively backward in economic development, far from the large-scale tourist market, with the primary and secondary industries as the leading industries, and the cultural tourism industry chain is not significant, belonging to the regions at the early stage of cultural tourism development. If the cultural tourism resources are rich, it may develop into a medium potential area.

The CTDP in the research area needs to be improved. There are significant spatial differences in the potential intensity of CTDP in the research area based on the mean. Most of the cities correspond to a level III area. Although the levels of CTDP in all cities are gradually improving, there are too few areas with high potential and too many areas with low potential. This does not match the basic conditions, social and economic environment and market demand of cultural tourism in the research area, so it is urgent to improve the CTDP.

With the great influence of time, the spatial correlation of urban CTDP is significant, and the level I areas are both concentrated and discrete. Time has a great influence on the CTDP in the research area, resulting in the fluctuation of the CTDP in most cities on the temporal axis. However, Xi'an and Qingdao are less affected by time, and their growth along the temporal axis is relatively stable. The CTDP in most cities is highly dependent on the region and greatly influenced by other cities in the research area. This is consistent with the spatially correlated characteristics of leisure and tourism facilities in Xi'an [72]. From the perspective of spatial correlation and spatial interaction, except for Xi'an and Qingdao, which have weak correlation with the CTDP in the surrounding cities, other cities have significant spatial correlation with the surrounding cities, which indicates that the development of urban cultural tourism has strong short-range regional dependence. The CTDP in most cities is greatly influenced by other cities, showing significant regional spatial interaction, which indicates that the research area has significant basic advantages in regional coordination, regional cooperation and overall development of cultural tourism development.

The spatial movement of the potential centroid tends to the geometric center, and the region through which its movement track passes is the area with the most concentrated distribution of traditional villages [40]. The spatiotemporal pattern of the CTDP centroid shows characteristics of fluctuation transferring towards the middle reaches of the Yellow River. Due to the significant difference in the time series of transferring distance and deviation angle, the spatial relationship of cultural tourism development among cities in different time periods is significantly different.

*5.2. Suggestions*

In order to enhance the CTDP of cities in the middle and lower reaches of the Yellow River, the following suggestions are put forward:

First, the development of cultural tourism not only needs the support of cultural tourism attractions and tourism reception facilities, but it also needs the support of regional investment and financing, scientific and technological systems, the cultural system, the talent system, Internet facilities and other resources. Other cradles of world civilization

like the middle and lower reaches of the Yellow River are also rich in high-quality cultural tourism resources. In the process of modernization and rapid economic development, sustainable development should be the main concept of the development in these areas, but the CTDP in these areas is relatively insufficient. Therefore, this study believes that the development of cultural tourism in the cradles of world civilization including the research area needs to implement the concept of sustainable development, integrate Sustainable Development Goal 12 SCP (Goal 12: ensure sustainable consumption and production patterns) into tourism policies, motivate tourism stakeholders, build a new pattern of cultural tourism development and improve the resource efficiency of cultural tourism. For the middle and lower reaches of the Yellow River, the cultural tourism development resources of various cities should be allocated as a whole, such as the technology system, the cultural system, the cultural tourism talent structure and the enterprise operation and management system, contributing to realizing coordination and linkage of transportation facilities, Internet facilities and online platforms. The Yellow River cultural resources should be shared, and it is necessary to promote the accessibility of transportation, the integration of technology, the coordination of systems and the flow of high-end talents within the region. For instance, realizing the sharing of excellent tour guides could better allocate potential resources for regional cultural tourism development [73].

Second, in regional development, cities have significant functions of factor aggregation, spatial radiation, development driving and innovation. The promotion of CTDP in the cradles of world civilization depends on cities' full implementation of cultural tourism potential. This study finds that urban CTDP in the research area is significantly different and low on the whole, which is consistent with the view of the development potential of small and medium-sized traditional villages and cultural heritage tourism within the birthplace of world civilization [24,44,53,74,75]. Therefore, this study argues that it is essential to enhance urban CTDP, which has three points: (1) relevant policies and systems should be formulated to increase patent development of smart cultural tourism, so as to strengthen urban technology innovation; (2) the supply factors of cultural tourism should be optimized, and the contribution capacity of the cultural tourism industry should be enhanced to improve its economic benefits; (3) tourist source markets should be opened up, and consumption capacity of urban cultural tourism should be enhanced.

Third, there is a large gap between the high potential area of cultural tourism development and the surrounding area, and there is a significant trough zone, with strong short-range regional dependence. Therefore, further strengthening cooperation and coordination within the region can effectively enhance the development potential of regional cultural tourism and better promote the implementation of the regional sustainable development strategy. Thus, this study prompts three practical suggestions: (1) Strengthen the leading role of the cities with the greatest regional development potential, give full play to their competition in cultural tourism development and vigorously promote the sustainable development of cultural tourism. Xi'an, as a high-potential area, has the weakest correlation with its surrounding areas, so it is advisable to promote the coordinated development of cultural tourism among its related cities with the help of the establishment of the urban agglomeration mechanism. (2) Core cities in the region are greatly affected by changes in the CTDP of other cities to a certain extent. In strengthening the development connection with surrounding cities, these cities should jointly develop a cultural tourism information system platform with intelligent technologies to achieve cooperation and coordinated development. At the same time, strengthen the primacy of central cities and enhance their spatial radiation capacity of cultural tourism development [61]. (3) According to the characteristics of significant spatial correlation of CTDP found in this study, cooperation and coordinated development among surrounding cities can be implemented. It can attract consumers and expand the scale of the consumer market by connecting the near-distance urban agglomeration with a high-tech metaverse. For example, the increase in the number of tourists and the healthy and active tourism demand can promote the development of urban cultural tourism [76,77]. Integrate the supply factors of cultural tourism resources

in surrounding cities, improve the supply chain, extend the industrial chain, improve the quality of service and implement a coordinated high-quality development strategy. For example, the European Union's regional coordinated development strategy has significantly promoted the development of cultural tourism in Italy [78].

*5.3. Discussions*

The basins where world civilization originated are cultural tourism destinations with abundant resources, high development value and great market potential in cultural tourism. However, they have received little attention from the academic circle. As one of the cradles of world civilization, the Yellow River Basin has formed local cultural systems and different cultural tourism products in its continuous cultural inheritance and promotion, highlighting the differences in regional and national governance of cultural tourism development. This will have the same goal as the European strategy for the development of smart cultural tourism, which is to promote cultural tourism as a driving force for sustainable regional development [79]. Evaluating the development potential of cultural tourism through GIS and TOPSIS is of great significance for the sustainable development of regional cultural tourism [80]. Therefore, on the basis of measuring the CTDP in the middle and lower reaches of the Yellow River, this paper innovatively discusses its spatiotemporal differentiation and puts forward a differentiated development strategy. Although the existing methods for evaluating tourism development potential lack comprehensibility, dynamics and visualization, this paper tries to innovate by collecting time series data and visualizing the evolution characteristics of the CTDP in the Yellow River based on the principle of multiple linear functions. However, this paper still has some limitations: (1) Only the middle and lower reaches of the Yellow River are selected in this paper to explore the CTDP in the birthplaces of world civilization. The potential level, the spatial pattern of potential intensity and the spatial aggregation characteristics of potential intensity obtained from this study must be further tested for universality. (2) The selection of evaluation indicators in this study is only based on the existing research results, and there is a lack of diversified selection methods, such as selecting evaluation indicators through expert consultation [81]. (3) The discussion on the attribution of influencing factors is limited. In the future, the factors affecting the spatiotemporal differentiation of CTDP and their attribution should be explored with the help of Geodetector.

**Author Contributions:** Conceptualization, Yuying Chen, Qi Jin and Qing Yuan; methodology, Yajie Li, Xiangfeng Gu and Qing Yuan; software, Xiangfeng Gu and Yajie Li; validation, Yajie Li, Xiangfeng Gu and Nan Chen; formal analysis, Yuying Chen, Yajie Li, Xiangfeng Gu, Qi Jin, Nan Chen and Qing Yuan; investigation, Yuying Chen and Yajie Li; resources, Yuying Chen, Yajie Li and Nan Chen; data curation, Yuying Chen, Yajie Li and Qing Yuan; writing—original draft preparation, Yuying Chen, Yajie Li and Xiangfeng Gu; writing—review and editing, Qi Jin, Qing Yuan and Nan Chen; visualization, Xiangfeng Gu; supervision, Yuying Chen, Qi Jin and Nan Chen; project administration, Nan Chen and Qi Jin; funding acquisition, Yuying Chen, Nan Chen and Qing Yuan. All authors have read and agreed to the published version of the manuscript.

**Funding:** This research was funded by the National Natural Science Foundation of China, grant number 42171186; Henan Provincial Department of Science and Technology (Soft Science), grant numbers 222400410493 and 232400410103; China National Social Science Fund, grant number 19BGJ007; China National Social Science Fund in Art, grant number 21ZD03; Ministry of Education of the People's Republic of China, grant number 22YJC760118; Key Scientific Research Project of Henan Province Colleges and Universities, grant number 23A790004.

**Data Availability Statement:** Data sharing not applicable.

**Acknowledgments:** The authors thank graduate and undergraduate students for their help during the data collection process: Meiting Liu, Xinling Wang, Jingjing Wang, Bohui Zhao, Jianchong Yu, Chaoyi Si and Ziyao Yin. The authors would also like to thank the editor and the anonymous referee of this journal for fruitful comments and suggestions that enhanced the paper's merit. All errors belong to the authors. The usual disclaimer applies.

**Conflicts of Interest:** The authors declare no conflict of interest.

**Appendix A**

Appendix A presents a detailed attribution of the original data sources for this article. Among them, the statistical yearbooks and statistical bulletins of each province are illustrated by taking Henan Province as an example, and the statistical yearbooks and statistical bulletins of each city are illustrated by taking Zhengzhou City as an example.

**Table A1.** Indicator sources.

| Indicators | Data Source | Data Access |
|---|---|---|
| highway passenger volume ($X_{111}$) | Chinese urban Statistical Yearbook; Henan statistical yearbook ... | https://data.cnki.net/yearBook/single?id=N2023070131 (2023-04-10) https://www.henan.gov.cn/zwgk/zfxxgk/fdzdgknr/tjxx/tjnj/ (2023-04-10) …..… |
| civil aviation passenger volume ($X_{112}$) | Zhengzhou Statistical Yearbook ... | https://data.cnki.net/yearBook/single?id=N2023030129 (2023-04-10) …..… |
| per capita urban road area ($X_{113}$) | Statistical yearbook of Chinese urban construction; Zhengzhou Statistical Yearbook ... | https://www.mohurd.gov.cn/gongkai/fdzdgknr/sjfb/tjxx/index.html (2023-04-10) https://data.cnki.net/yearBook/single?id=N2023030129 (2023-04-10) …..… |
| harmless treatment rate of household garbage ($X_{114}$) | Chinese urban Statistical Yearbook; Statistical yearbook of Chinese urban construction ... | https://data.cnki.net/yearBook/single?id=N2023070131 (2023-04-10) https://www.mohurd.gov.cn/gongkai/fdzdgknr/sjfb/tjxx/index.html (2023-04-10) …..… |
| Total telecommunications services ($X_{115}$) | Henan statistical yearbook; Zhengzhou Statistical Yearbook ... | https://www.henan.gov.cn/zwgk/zfxxgk/fdzdgknr/tjxx/tjnj/ (2023-04-10) https://data.cnki.net/yearBook/single?id=N2023030129 (2023-04-10) …..… |
| Days with good air quality ($X_{121}$) | Zhengzhou National economic and social development Bulletin ... | https://tjj.zhengzhou.gov.cn/tjgb/index.jhtml (2023-04-10) …..… |
| Green coverage in built-up areas ($X_{122}$) | Chinese urban Statistical Yearbook; Statistical yearbook of Chinese urban construction ... | https://data.cnki.net/yearBook/single?id=N2023070131 (2023-04-10) https://www.mohurd.gov.cn/gongkai/fdzdgknr/sjfb/tjxx/index.html (2023-04-10) …..… |
| park land per capita ($X_{123}$) | Chinese urban Statistical Yearbook; Statistical yearbook of Chinese urban construction ... | https://data.cnki.net/yearBook/single?id=N2023070131 (2023-04-10) https://www.mohurd.gov.cn/gongkai/fdzdgknr/sjfb/tjxx/index.html (2023-04-10) …..… |

**Table A1.** *Cont.*

| Indicators | Data Source | Data Access |
|---|---|---|
| Regional population density ($X_{124}$) | Statistical yearbook of Chinese urban construction; Zhengzhou Statistical Yearbook ... | https://www.mohurd.gov.cn/gongkai/fdzdgknr/sjfb/tjxx/index.html (2023-04-10) https://data.cnki.net/yearBook/single?id=N2023030129 (2023-04-10) ...... |
| proportion of the tertiary industry in the gross regional product ($X_{131}$) | Chinese urban Statistical Yearbook | https://data.cnki.net/yearBook/single?id=N2023070131 (2023-04-10) |
| proportion of fixed investment in the tertiary industry ($X_{132}$) | Henan statistical yearbook; Zhengzhou National Economic and social Development Bulletin ... | https://www.henan.gov.cn/zwgk/zfxxgk/fdzdgknr/tjxx/tjnj/ (2023-04-10) https://tjj.zhengzhou.gov.cn/tjgb/index.jhtml (2023-04-10) ...... |
| total amount of actually utilized foreign investment ($X_{133}$) | Zhengzhou Statistical Yearbook ... | https://data.cnki.net/yearBook/single?id=N2023030129 (2023-04-10) ...... |
| fixed assets investment in municipal public facilities construction ($X_{134}$) | Statistical yearbook of Chinese urban construction; Zhengzhou Statistical Yearbook ... | https://www.mohurd.gov.cn/gongkai/fdzdgknr/sjfb/tjxx/jstjnj/index.html (2023-04-10) https://data.cnki.net/yearBook/single?id=N2023030129 (2023-04-10) ...... |
| urbanization rate ($X_{135}$) | Zhengzhou Statistical Yearbook ... | https://data.cnki.net/yearBook/single?id=N2023030129 (2023-04-10) ...... |
| domestic tourist arrivals ($X_{211}$) | Zhengzhou Statistical Yearbook ... | https://data.cnki.net/yearBook/single?id=N2023030129 (2023-04-10) ...... |
| inbound tourist arrivals ($X_{212}$) | Zhengzhou Statistical Yearbook ... | https://data.cnki.net/yearBook/single?id=N2023030129 (2023-04-10) ...... |
| Engel coefficient ($X_{221}$) | Zhengzhou Statistical Yearbook ... | https://data.cnki.net/yearBook/single?id=N2023030129 (2023-04-10) ...... |
| per capita GDP ($X_{222}$) | Chinese urban Statistical Yearbook | https://data.cnki.net/yearBook/single?id=N2023070131 (2023-04-10) |
| Per capita expenditure on education, culture and entertainment ($X_{223}$) | Henan statistical yearbook; Zhengzhou Statistical Yearbook ... | https://www.henan.gov.cn/zwgk/zfxxgk/fdzdgknr/tjxx/tjnj/ (2023-04-10) https://data.cnki.net/yearBook/single?id=N2023030129 (2023-04-10) ...... |
| Number of high A-level scenic spots ($X_{311}$) | Zhengzhou National economic and social development Bulletin ... | https://tjj.zhengzhou.gov.cn/tjgb/index.jhtml (2023-04-10) ...... |
| Number of enterprises in the accommodation and catering industry ($X_{312}$) | Zhengzhou Statistical Yearbook ... | https://data.cnki.net/yearBook/single?id=N2023030129 (2023-04-10) ...... |
| Number of public toilets ($X_{313}$) | Zhengzhou Statistical Yearbook ... | https://data.cnki.net/yearBook/single?id=N2023030129 (2023-04-10) ...... |
| Number of cultural heritage ($X_{314}$) | Government data of the Ministry of Culture and Tourism, PRC; | https://sjfw.mct.gov.cn/site/dataservice/culture (2023-04-10) |

**Table A1.** *Cont.*

| Indicators | Data Source | Data Access |
|---|---|---|
| Number of cultural institutions ($X_{315}$) | Henan statistical yearbook Zhengzhou Statistical Yearbook ... | https://www.henan.gov.cn/zwgk/zfxxgk/fdzdgknr/tjxx/tjnj/ (2023-04-10) https://data.cnki.net/yearBook/single?id=N2023030129 (2023-04-10) ...... |
| Total tourism revenue ($X_{321}$) | Zhengzhou Statistical Yearbook ... | https://data.cnki.net/yearBook/single?id=N2023030129 (2023-04-10) ...... |
| Tourism foreign exchange revenue ($X_{322}$) | Zhengzhou Statistical Yearbook ... | https://data.cnki.net/yearBook/single?id=N2023030129 (2023-04-10) ...... |
| Proportion of employment in the tertiary industry ($X_{323}$) | Henan statistical yearbook Zhengzhou Statistical Yearbook ... | https://www.henan.gov.cn/zwgk/zfxxgk/fdzdgknr/tjxx/tjnj/ (2023-04-10) https://data.cnki.net/yearBook/single?id=N2023030129 (2023-04-10) ...... |
| Number of employees in culture, sports and entertainment industry ($X_{324}$) | Henan statistical yearbook; Zhengzhou Statistical Yearbook ... | https://www.henan.gov.cn/zwgk/zfxxgk/fdzdgknr/tjxx/tjnj/ (2023-04-10) https://data.cnki.net/yearBook/single?id=N2023030129 (2023-04-10) ...... |
| Number of employees in the accommodation and catering industry ($X_{325}$) | Zhengzhou Statistical Yearbook ... | https://data.cnki.net/yearBook/single?id=N2023030129 (2023-04-10) ...... |
| Number of R&D personnel ($X_{411}$) | Henan statistical yearbook; Zhengzhou Statistical Yearbook ... | https://www.henan.gov.cn/zwgk/zfxxgk/fdzdgknr/tjxx/tjnj/ (2023-04-10) https://data.cnki.net/yearBook/single?id=N2023030129 (2023-04-10) ...... |
| R&D expenditure ($X_{412}$) | Henan statistical yearbook; Zhengzhou Statistical Yearbook ... | https://www.henan.gov.cn/zwgk/zfxxgk/fdzdgknr/tjxx/tjnj/ (2023-04-10) https://data.cnki.net/yearBook/single?id=N2023030129 (2023-04-10) ...... |
| Number of patent grants ($X_{413}$) | Chinese urban Statistical Yearbook ... | https://data.cnki.net/yearBook/single?id=N2023070131 (2023-04-10) ...... |

Data source: collated by the author.

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
