# Peer review of "Evaluation and Spatiotemporal Differentiation of Cultural Tourism Development Potential: The Case of the Middle and Lower Reaches of the Yellow River"

_ijgi, doi:10.3390/ijgi12110461_

Round 1

Reviewer 1 Report

Comments and Suggestions for Authors

Review report for IJGI

Article code: ijgi-2633255

  Evaluation and Spatio-temporal Differentiation of Cultural Tourism Development Potential: The Case of the Middle and Lower Reaches of the Yellow River

Comments and Revisions: 

Introduction

- This section suffers from a compelling argument on the main subject of the study. It has lost its concentration on the most important issue of the study (cultural tourism, regional sustainable development, and spatial analysis approach). 

- There are some vague sentences that I cannot follow (e.g. lines 42 and 43)

- There are some statements without being cited (e.g. lines45-50)

- The contributions of the present study to the concurrent literature have been overlooked totally.

-Materials and Methods

-  The lack of scientific justification for the assessing variables and factors makes them less reliable. Therefore, it is essential to first justify the assessing variables through domestic and international literature.

-  The judgment system articulated in Table 1 is based on only one person which is somehow vague and unacceptable. Moreover, the judging system needs to be explained in detail.

- The quality of the presented maps is very low.

- How 5 levels of CTDP score has been calculated? and in what scientific base each level has been classified? Is there any literature support? 

_ Fig.1 needs to be revised so that it shows firstly the country and then the study area

Discussion

- One of the main important issues in the study has been neglected in this section; in fact, this question was supposed to be addressed why did some cities in the study area gain higher scores over time? 

- There is a lack of comparative study in this part of the study and I recommend authors review the following articles and compare their findings with others.

1-Pazhuhan, M. and Shiri, N. (2020), "Regional tourism axes identification using GIS and TOPSIS model (Case study: Hormozgan Province, Iran)", Journal of Tourism Analysis: Revista de Análisis Turístico, Vol. 27 No. 2, pp. 119-141. https://doi.org/10.1108/JTA-06-2019-0024

2- Identifying the Spatial Patterns and Influencing Factors of Leisure and Tourism in Xi’an Based on Point of Interest (POI) Data-Land-2023, 12, 1805.

3- Michálková, A., Krošláková, M.N., ÄŒvirik, M. et al. Analysis of management on the development of regional tourism in Europe. Int Entrep Manag J 19, 733–754 (2023). https://doi.org/10.1007/s11365-023-00840-x

4- Smart Cultural Tourism as a Driver of Sustainable Development of European Regions – SmartCulTour, UNESCO

Comments on the Quality of English Language

Moderate editing of English language required

Author Response

Article code: ijgi-2633255

Evaluation and Spatio-temporal Differentiation of Cultural Tourism Development Potential: The Case of the Middle and Lower Reaches of the Yellow River

Response to Reviewer #1

Dear Reviewer,

Thank you so much for your comments. The comments you have proposed for us are all valuable and very helpful for revising and improving our paper, as well as the essential guiding significance to our study. We have studied the comments carefully and made some corrections, which we hope to meet with approval. The revised portion is marked in red on the paper. Furthermore, the response to your comments is as follows:

Comment 1:

(Introduction)- This section suffers from a compelling argument on the main subject of the study. It has lost its concentration on the most important issue of the study (cultural tourism, regional sustainable development, and spatial analysis approach).

Response 1:

We agree with the reviewers that some essential questions from the study are missing from the introduction. Therefore, we removed and modified some things based on the reviewers' recommendations, as follows:

On page 2, lines 53-63, we have added some descriptions of the meaning, content, and research theme of cultural tourism. “The study of cultural tourism began in the United States in the fifties of the last century; scholars believe that cultural tourism is to meet the cultural needs of tourists and cause tourists to think about social and human aspects of tourism activities, including handicrafts, art, and music, architecture, perception of tourist destinations, monuments, festivals, heritage resources, technology, religion, education, and other contents [7]. At present, the research on cultural tourism mainly focuses on cultural tourism experience, cultural tourism authenticity, and tourism development of cultural heritage [8,9]. There are some studies on the tourism development potential of cultural heritage, and there are almost no studies on CTDP. Still, the cultural significance has increasingly become an important dimension in assessing regional tourism development potential [4]."

On page 2, delete line 50, "Also, in the 21st century, Chinese scholars began to discuss the relationship between culture and tourism."

On page 2, lines 82-85, an explanation of the spatial analysis method is added." Based on a comprehensive potential assessment, this paper combines geographic information systems (GIS) for in-depth analysis. It mainly uses cluster analysis, Spatial kernel density estimation, Centroid transfer model, and Kringing interpolation analysis."

Comment 2:

(Introduction)- There are some vague sentences that I cannot follow (e.g. lines 42 and 43)

Response 2:

On page 2, line 45, The sentence “The cultural gene of the Yellow River should be deeply inherited, and a Yellow River cultural tourism belt with international influence should be established.” comes from the “Outline of the Yellow River Basin's Ecological Protection and High-quality Development Plan” issued by the Chinese government department. It is a strategic deployment made by China to protect, inherit, and promote the Yellow River culture. In order to make the sentence more straightforward, delete the phrase "The cultural gene of the Yellow River should be deeply inherited,"

Comment 3:

(Introduction)- There are some statements without being cited (e.g., lines45-50)

Response 3:

We apologize for missing some reference citations here, and thanks to the reviewers' reminders and suggestions, we have cited and annotated the literature, documents, and regulations that appear in the introduction of the article and added the references as follows:

Added reference [1] “Towner, J. WHAT IS TOURISMS HISTORY. Tourism Manage. 1995, 16, 339-343, doi:10.1016/0261-5177(95)00032-j.” to the sentence “As early as the 16th century, The Grand tour put a premium on exotic cultural experiences.” of 47-48 lines.

Added reference [2] “UNWTO. Global Code of Ethics for Tourism. Available online: https://www.unwto.org/global-code-of-ethics-for-tourism (accessed on 2023-10-28).).” to the sentence “In 1999, UNWTO published “Global Code of Ethics for Tourism,” which emphasizes that tourism should be developed under the premise of protecting culture.” of lines 48-49.

Comment 4:

(Introduction)- The contributions of the present study to the concurrent literature have been overlooked totally.

Response 4:

We agree with the reviewers. According to the reviewer's recommendations, on page 2, lines 94-97, “in order to provide decision-making reference for the high-quality and sustainable development of cultural tourism in the river basin.” was revised as “From the perspective of potential, it enriches the research on cultural tourism, regional sustainable development, and provides decision-making reference for the high-quality and sustainable development of cultural tourism in the basin where world civilization originated.”

Comment 5:

(Materials and Methods)- The lack of scientific justification for the assessing variables and factors makes them less reliable. Therefore, it is essential to first justify the assessing variables through domestic and international literature.

Response 5:

To justify and verify the evaluation variables, we supplemented the literature review. We added a research review on the index system of cultural tourism development potential: " In general, the evaluation index system that can be used for reference in CTDP evaluation is mainly constructed from the influencing factors and market players. From the analysis of the influencing factors of development potential, scholars pay attention to its internal and external factors, that is, the self-development potential of the tourism industry (internal resources and benefits) and the external guarantee and support power (infrastructure, services, and environment) [31-33]. From the perspective of tourism market players, scholars construct an index system from the market demand and purchase of the supply and demand side of tourism factors (attraction and cultural tourism enterprises) [13,34-36]. In terms of the evaluation indicators of CTDP, scholars mainly use indicators such as cultural tourism consumption capacity, operating accommodation capacity, number of tourists, number of overnight stays, average stay time, and public and private sector investment in cultural tourism projects [37-39] to evaluate from multiple perspectives such as regional tourism planning, heritage protection strategy formulation, and cultural tourism industry."

Comment 6:

(Materials and Methods)- The judgment system articulated in Table 1 is based on only one person which is somehow vague and unacceptable. Moreover, the judging system needs to be explained in detail.

Response 6:

First, we have supplemented the 2.2 literature review section and added a research review of the index system of cultural tourism development potential.

Then, we have described the source of the indicator system in the 3.1 Theoretical framework, that is, “According to the evaluation index system of the tourism development potential of provinces along the Belt and Road in China built by Yuying Chen (the first author) [13]. Then, it refers to the relevant evaluation indexes in the existing research achievements of cultural development potential, tourism development potential, and heritage tourism development potential [28,34,37-41]”.

The detailed indicator description is shown in 3.1.1-3.1.4.

Comment 7:

(Materials and Methods)- The quality of the presented maps is very low.

Response 7:

Sorry for the poor reading experience for reviewers due to the low quality of the map. We have readjusted the arrangement of the map and made the image clear.

Comment 8:

(Materials and Methods)- How 5 levels of CTDP score has been calculated? and in what scientific base each level has been classified? Is there any literature support?

Response 8:

On page 13, lines 439-444, “According to the Table 2, using SPSS statistical cluster analysis[55], it is found that the CTDP score of 43 cities can be clustered into five levels,” was revised as “According to the Table 2, this paper adopts k-means clustering method for cluster analysis with the help of SPSS software. In general, the value of K is 3-8[33,34,65-67,69]. Due to the large number of objects studied in this paper, this paper sets K as five and then adopts an iterative method according to the minimum Euclidean distance from the class center to better explore the hierarchical spatial structure characteristics. By measuring its clustering center, the CTDP score of 43 cities can be clustered into five levels [65,66,70,71],”

Comment 9:

(Materials and Methods)- Fig.1 needs to be revised so that it shows the country and then the study area

Response 9:

Thanks to the comments of the reviewers. As an international journal, this suggestion is necessary. We modified the original Figure 1 to make a location map to clearly and visually show the location and scope of the research area in China as shown in Figure 1.

Comment 10:

(Discussion)- One of the main important issues in the study has been neglected in this section; in fact, this question was supposed to be addressed why did some cities in the study area gain higher scores over time?

Response 10:

It is vital to explain each conclusion. In response to the comments raised by the reviewers, this article made some changes to explain why a city's CTDP score has increased over time.

On page 21, lines 628-635, Removed " According to the mean of the potential score, the CTDP of all cities is low on the whole,…, which does not match the resource base of high concentration of the Yellow River civilization symbols and carriers." and added " Due to the social and economic development and scientific and technological progress in the Yellow River Basin, the public environmental supply of cities has become complete, the quality of the natural environment has been continuously improved, the economic support capacity has gradually increased, the demand and purchasing power of tourists has increased, and the supply of cultural tourism industry has met the demand. Its scale and benefits have been continuously improved. The society has paid more and more attention to scientific and technological innovation, and the development of cultural tourism has become more intelligent."

Comment 11:

(Discussion)- There is a lack of comparative study in this part of the study and I recommend authors review the following articles and compare their findings with others.

Response 11:

Sincere thanks to the reviewers for their suggestions, as well as for the literature that took the time to provide. We carefully read the literature provided by the reviewers and compared and analyzed it concerning this manuscript, and the specific revisions are as follows:

Through reading and thinking about the first article ( [80] Pazhuhan, M.; Shiri, N. Regional tourism axes identification using GIS and TOPSIS model (Case study: Hormozgan Province, Iran). J. Tour. Anal. 2020, 27, 119-141, doi:https://doi.org/10.1108/JTA-06-2019-0024), we find that it has similarities with this paper in terms of research methodology, that is, “It is significant to evaluate the development potential of cultural tourism through GIS and TOPSIS, and can better implement the sustainable.” Development of regional cultural tourism.”

The second article ( [72] Qu, X.S.; Xu, G.Y.; Qi, J.H.; Bao, H.J. Identifying the Spatial Patterns and Influencing Factors of Leisure and Tourism in Xi’an Based on Point of Interest (POI) Data. Land 2023, 12, 18, doi:10.3390/land12091805.) has some similarities with the conclusion of this paper: "With the great influence of time, the spatial correlation of urban CTDP is significant," that is, "This is consistent with the spatially correlated characteristics of." leisure and tourism facilities in Xi 'an.”

The selection of evaluation indicators in the third article ( [81] Michálková, A.; Kroslakova, M.N.; Cvirik, M.; Martínez, J.M.G. Analysis of management on the development of regional tourism in Europe. Int. Entrep. Manag. J. 2023, 19, 733-754, doi:10.1007/s11365-023-00840-x.) has led to some thoughts about the lack of research, that is, “The selection of evaluation indicators in this study is only based on the existing research results, and there is a lack of diversified selection methods, such as selecting evaluation indicators through expert consultation.”

The fourth document ([79] UNESCO. Smart Cultural Tourism as a Driver of Sustainable Development of European Regions - SmartCulTour. Available online: https://www.unesco.org/en/articles/smart-cultural-tourism-driver-sustainable-development-european-regions-smartcultour (accessed on 2023-10-28)) recommended by the reviewer is consistent with this paper in terms of research objectives and conclusions, that is, "This will have the same goal as the European strategy for the development of smart cultural tourism, which is to promote cultural tourism as a driving force for sustainable regional development.”

Comment 12:

Moderate editing of English language required.

Response 12:

Thank you for the valuable and thoughtful comments. We have carefully checked and improved the English writing in the revised manuscript.

We would like to take this opportunity to thank you for all the time and valuable suggestions for us to improve the manuscript. We hope the revised version will find you well.

Sincerely,

The Authors

Reviewer 2 Report

Comments and Suggestions for Authors

The paper effectively highlights the significant potential for cultural tourism in the Middle and Lower Reaches of the Yellow River. Despite being robust, the article has some issues with its structure and references. Therefore, there are opportunities to enhance the overall quality of the paper by addressing the following suggestions:

1 - Abstract Clarity: The abstract should include more information about the rationale, objectives, and methodology of the study to provide readers with a clearer understanding of its scope. Additionally, it should better reflect the extensive literature review conducted on CTDP.

2 - Introduction References: The historical overview in the introduction should be substantiated with appropriate references, such as articles, books, documents, or legislation, to strengthen the foundation of the study.

3 - Global Context: In the introduction, it would be valuable to mention relevant studies conducted in other countries with similar objectives. This could provide insight into methodological alternatives and broaden the article's international context.

4 - Reference Map: Consider including a reference map in Figure 1 to help foreign readers locate the study area within China.

5 - Data and References: In section 2.3, it is crucial to reference the documents and datasets mentioned. Additionally, presenting these data in a table format would enhance their visibility and accessibility.

6 - Equation References: Ensure that all equations mentioned in the text are appropriately referenced.

7 - Section Clarity: Clearly distinguish between the literature review, methodology, and results sections to improve the article's organization and readability.

8 – Discussion Focus: There is misunderstanding in the discussion section where the results are explored repeatedly. I believe that the discussion section should focus on explaining the findings, highlighting their implications and potential alternatives. Furthermore, in the discussion, the suggested alternatives for the study area need to be supported by other studies (references) that have taken similar actions or literature reviews that recommend such actions.

9 - Conclusion: The article would benefit from a well-structured conclusion to succinctly summarize key findings and their significance.

Author Response

Response to Reviewer #2

Dear Reviewer,

Thanks very much for taking the time to review this manuscript. We really appreciate all your comments and suggestions! We have studied the comments carefully and made corrections to each one. The revised portion is marked in red in the manuscript. The response to your comments is as follows:

Comment 1:

1 - Abstract Clarity: The abstract should include more information about the rationale, objectives, and methodology of the study to provide readers with a clearer understanding of its scope. Additionally, it should better reflect the extensive literature review conducted on CTDP.

Response 1:

We adjusted the abstract section to clarify the rationale, objectives, and methodology of the study and added some literature review to make the abstract section more complete, as follows:

On page 1, lines 15-17, "The existing methods of evaluating development potential lack dynamics and visualization.” was revised as “At present, there are few relevant studies on CTDP, but the research results on the tourism development potential of cultural heritage are relatively rich, and the existing evaluation methods lack comprehensiveness, dynamics, and visualization.”

On page 1, lines 17-22, “This paper, through collecting time series data and constructing an evaluation model, makes an innovative attempt to use dynamic kernel density and centroid transferring curve to visualize the spatio-temporal evolution characteristics of the CTDP of 43 cities in the research area." was revised as " Based on Systems Theory and Sustainable Development Theory, this manuscript attempts to innovate and collect time series data through Entropy Method, Multi-index Comprehensive Evaluation Method, Spatial Kernel Density Estimation Method, and Centroid Transferring Model. The temporal and spatial evolution characteristics and the CTDP of 43 cities in the middle and lower reaches of the Yellow River are examined and analyzed."

On page 1, lines 22-23, "It is found that the CTDP is divided into five levels." was revised as "It was found that CTDP in the middle and lower reaches of the Yellow River was divided into five levels."

Comment 2:

2 - Introduction References: The historical overview in the introduction should be substantiated with appropriate references, such as articles, books, documents, or legislation, to strengthen the foundation of the study.

Response 2:

We apologize for missing some reference citations here, and thanks to the reviewers' reminders and suggestions, we have cited and annotated the literature, documents, and regulations that appear in the introduction of the article and added the references as follows:

Added reference [1] “Towner, J. WHAT IS TOURISMS HISTORY. Tourism Manage. 1995, 16, 339-343, doi:10.1016/0261-5177(95)00032-j.” to the sentence “As early as the 16th century, The Grand tour put a premium on exotic cultural experiences.” of 47-48 lines.

Added reference [2] “UNWTO. Global Code of Ethics for Tourism. Available online: https://www.unwto.org/global-code-of-ethics-for-tourism (accessed on 2023-10-28).).” to the sentence “In 1999, UNWTO published “Global Code of Ethics for Tourism,” which emphasizes that tourism should be developed under the premise of protecting culture.” of lines 48-49.

Comment 3:

3 - Global Context: In the introduction, it would be valuable to mention relevant studies conducted in other countries with similar objectives. This could provide insight into methodological alternatives and broaden the article's international context.

Response 3:

First, we mentioned the Nile, Euphrates, Tigris, Indus, and Yellow River, the birthplaces of world civilization throughout history, in the first sentence of the introduction.

Then, in the last paragraph, we added an international practical significance, that is, "It also provides a reference for other river basin civilizations in the world."

In Article 5 Conclusions and discussions, this study added a summary and discussion of the Yellow River as the birthplace of world civilization, that is, "The birthplaces of world civilization are cultural tourism destinations with abundant resources, high development value and great market potential in cultural tourism. As cultural tourism destinations on the macro-mesoscale, they have received little attention from the academic circle. As one of the cradles of world civilization, the Yellow River has formed local cultural systems and different cultural tourism products in its continuous cultural inheritance and promotion, highlighting the differences in regional and national governance of cultural tourism development. It will have the same goal as the European strategy for developing smart cultural tourism, which is to promote cultural tourism as a driving force for sustainable regional development [79]."

Comment 4:

4 - Reference Map: Consider including a reference map in Figure 1 to help foreign readers locate the study area within China.

Response 4:

Thanks to the reviewers' comments, as an international journal, the suggestion is essential. We modified the original Figure 1 to make a location map to show the location and scope of the research area clearly and visually in China (as shown in Figure 1).

Comment 5:

5 - Data and References: In section 2.3, it is crucial to reference the documents and datasets mentioned. Additionally, presenting these data in a table format would enhance their visibility and accessibility.

Response 5:

We built a table to show the source of the metrics and how the data was obtained to enhance the visibility and accessibility of the data (as shown in Appendix A).

Comment 6:

6 - Equation References: Ensure all equations mentioned in the text are appropriately referenced.

Response 6:

We are very sorry that we have neglected to quote some equations, so we first modified the “2.4 Spatio-temporal differentiation measurement” to the “2.4 Research methods” and added the “2.4.1 Entropy method” section to explain the weight calculation method in detail [56,57]. The formula in the “2.4.3 Centroid shift model” was then supplemented with references to ensure proper citation [59].

Comment 7:

7 - Section Clarity: Clearly distinguish between the literature review, methodology, and results sections to improve the article's organization and readability.

Response 7:

We redefined the article structure, literature review, and research methodology in "2. Materials and Methods" is explained. Among them, 2.2 was a Literature Review, and 2.4 was research methods. The secondary headings in 4 Results have been adjusted to make the Results of this paper more legible: "4.1. CTDP score of cities in the middle and lower reaches of the Yellow River" was revised as "4.1. CTDP score of cities in the middle and lower reaches of the Yellow River continue to improve"; "4.2. Evolution of potential level of urban CTDP in the research area" was revised as "4.2. CTDP potential level of the cities in the study area is divided into five levels "; "4.3. Spatio-temporal differentiation of CTDP in the research area" was revised as "4.3. Spatio-temporal differentiation of CTDP in the research area is significant "; "4.4. Evolution characteristics of centroid transferring of CTDP in the research area" was revised as "4.4. Potential centroid of CTDP in the research area is gradually moving toward the geometric center ".

Comment 8:

8 – Discussion Focus: There is misunderstanding in the discussion section where the results are explored repeatedly. I believe that the discussion section should focus on explaining the findings, highlighting their implications and potential alternatives. Furthermore, in the discussion, the suggested alternatives for the study area need to be supported by other studies (references) that have taken similar actions or literature reviews that recommend such actions.

Response 8:

Thanks to the reviewers' comments, we have reorganized the discussion section and changed "5. Discussion" to "5. Conclusions and discussions." We set up a secondary heading to make it more transparent, that is, "5.1. Conclusions": Focus on the main conclusions of this paper and the reasons for the conclusions; "5.2. Suggestions": Based on the research conclusions of this paper, the opinions and suggestions on the improvement of cultural tourism potential and high-quality development in the Yellow River Basin are put forward; "5.3. Discussions" This part mainly explains the generality of the research results, significance, shortcomings, and prospects of this paper.

Comment 9:

9 - Conclusion: The article would benefit from a well-structured conclusion to succinctly summarize key findings and their significance.

Response 9:

As shown in Response 8, the article has changed the discussion section. The sentences with a certain degree of repetition with the research results were deleted. The research results are mainly summarized, and the research conclusions are formed with a focus on interpretation and meaning.

Lines 617-687 are the concluding sections of this paper, which summarize the six main conclusions and provide an in-depth analysis of each of them, including their meanings, causes, and theoretical implications.

We appreciate your warm work and valuable comments and hope that the correction we have carefully considered will meet with approval.

Sincerely,

The Authors

Reviewer 3 Report

Comments and Suggestions for Authors

The main idea of the article is to propose a new evaluation model for measuring the cultural tourism development potential in the Middle and Lower Reaches of the Yellow River in China

The article is about the evaluation of cultural tourism development potential in the Middle and Lower Reaches of the Yellow River in China. It proposes a quite old evaluation model that uses dynamic kernel density  and Centroid transferring model the spatio-temporal differentiation of cultural tourism development potential. The article also discusses the key factors that contribute to cultural tourism development potential, such as cultural tourism consumption capacity, operating accommodation capacity, number of tourists, investment of cultural tourism projects, and cultural significance.

All graphs are not suitable for publication, their size means they are unreadable. Especially Fig. 3 and 5, 6. Why are the tables unreadable, too big fonts.

Shouldn't the discussion section after the regression model state what the R2 (coefficient of determination) is and in 3.3. Perhaps the model is not correct and should not be used?

Author Response

Response to Reviewer #3

Dear Reviewers,

Thanks very much for taking the time to review this manuscript. The comments you have proposed for us are all valuable and very helpful for revising and improving our paper, as well as the essential guiding significance to our study. We have studied the comments carefully and made some corrections, which we hope to meet with approval. The revised portion is marked in red in the manuscript. The response to your comments is as follows:

Comment 1:

It proposes a quite old evaluation model that uses dynamic kernel density and Centroid transferring model the spatio-temporal differentiation of cultural tourism development potential.

Response 1:

Although there are currently many very novel and powerful models, the data span of this study is large, and the development of the case area is typical and contemporary. Using classical models can not only present the development effect most simply but also once again verify the adaptability of classical models in case areas of developing countries.

Comment 2:

All graphs are not suitable for publication, their size means they are unreadable. Especially Fig. 3 and 5, 6. Why are the tables unreadable, too big fonts.

Response 2:

Sorry for the poor reading experience for reviewers due to the low quality of the map. We have readjusted the arrangement of the map and made the image clear.

Comment 3:

Shouldn't the discussion section after the regression model state what the R2 (coefficient of determination) is and in 3.3. Perhaps the model is not correct and should not be used?

Response 3:

We have modified the description of the CTDP index model in 3.3 to explain the coefficients and variables of the model in detail.

On page 11, lines 398-404, “The basic principles of the evaluation model are as follows:” was revised as “The basic principle of this evaluation model is shown in Equation (4). By this principle, this paper firstly retrieves and processes the index data of each factor layer of 43 cities in the case area from 2009 to 2020 and then uses the entropy method to calculate the weights and scores step by step. The target layer scores can be obtained according to the weights and scores of the main criterion layer, that is, the comprehensive index of the cultural tourism development potential of each city, as shown in Equation (9).”

On page 11, lines 405-410, “?? is the comprehensive score of the i-th development potential evaluation rule layers, which is obtained by summing the original assignment of each factor layer contained in the i-th index after standardization by range method. ?? is the weight of the i-th rule layers. Through this principle, this paper applies the data assignment from 2009 to 2020 of 43 cities in the research area. According to the weight of rule layers obtained by mathematical statistics analysis, the evaluation model of CTDP of cities in the middle and lower reaches of the Yellow River can be obtained as follows:” was revised as “ is the weight of the i-th rule layers. As shown in Table 1, , ,  and .  is the comprehensive score of the i-th development potential evaluation rule layers. Among them,  is the environmental support potential,  is the demand potential of cultural tourism,  is the supply potential of cultural tourism,  is the technology innovation potential”.

Thank you very much for all your time involved and this great opportunity for us to improve the manuscript. We hope you will find this revised version satisfactory. We look forward to hearing from you.

Sincerely,

The Authors

Reviewer 4 Report

Comments and Suggestions for Authors

This paper, through collecting time series data and constructing an evaluation model supported by the multiple linear functions, makes an attempt to use spatial kernel density and centroid transferring curve to visualize the spatio-temporal evolution characteristics of the CTDP of 43 cities in the middle and lower reaches of the Yellow River.

The paper is relevant and of interest. There are several questions:

1. what is the degree of correlation for the CTDP index model for the middle and lower reaches of the Yellow River (6)?

2. What are the limitations for this multivariate model (6)?

Author Response

Response to Reviewer #4

Dear Reviewers,

Thanks very much for taking your comments and professional advice. These opinions help to improve the academic rigor of our article. Based on your suggestion and request, we have made corrected modifications to the revised manuscript. The modified portion is marked in red in the manuscript. Furthermore, we would like to show the details as follows:

Comment 1:

  1. What is the degree of correlation for the CTDP index model for the middle and lower reaches of the Yellow River (6)?

Response 1:

We have modified the description of the CTDP index model in 3.3 to explain the coefficients and variables of the model in detail.

On page 11, lines 398-404, “The basic principles of the evaluation model are as follows:” was revised as “The basic principle of this evaluation model is shown in Equation (4). By this principle, this paper firstly retrieves and processes the index data of each factor layer of 43 cities in the case area from 2009 to 2020 and then uses the entropy method to calculate the weights and scores step by step. The target layer scores can be obtained according to the weights and scores of the main criterion layer, that is, the comprehensive index of the cultural tourism development potential of each city, as shown in Equation (9).”

On page 11, lines 405-410, “?? is the comprehensive score of the i-th development potential evaluation rule layers, which is obtained by summing the original assignment of each factor layer contained in the i-th index after standardization by range method. ?? is the weight of the i-th rule layers. Through this principle, this paper applies the data assignment from 2009 to 2020 of 43 cities in the research area. According to the weight of rule layers obtained by mathematical statistics analysis, the evaluation model of CTDP of cities in the middle and lower reaches of the Yellow River can be obtained as follows:” was revised as “ is the weight of the i-th rule layers. As shown in Table 1, w1=0.2095, w2=0.1962, w3=0.2475, and w4=0.3486  is the comprehensive score of the i-th development potential evaluation rule layers. Among them,  x1 is the environmental support potential, x2 is the demand potential of cultural tourism, x3 is the supply potential of cultural tourism, x4 is the technology innovation potential”.

Comment 2:

  1. What are the limitations for this multivariate model (6)?

Response 2:

In conclusion, we added the scope of application of the model, that is, “the evaluation model of CTDP is only applicable to the study area, but the principle, method, and process of its model construction are universal and have reference significance for the study of the development potential of regional cultural tourism.”

We also illustrate the shortcomings of this study in the discussion section of 5.3. Discussions, “(1) Only the middle and lower reaches of the Yellow River are selected in this paper to explore the CTDP in the birthplaces of world civilization. The potential level, the spatial pattern of potential intensity, and spatial aggregation characteristics of potential intensity obtained from this study must be further tested for universality. (2) This study only analyzed the spatial aggregation characteristics of the CTDP of the cities in the research area. Hence, it needs to be followed up and supplemented through the analysis of spatio-temporal differentiation of potential in future research. (3) The discussion on the attribution of influencing factors is limited. In the future, the factors affecting the spatio-temporal differentiation of CTDP and their attribution should be explored with the help of Geodetector.

Once again, special thanks to you for all your good comments and suggestions. We hope this revised version will find you well.

Sincerely,

The Authors

Round 2

Reviewer 1 Report

Comments and Suggestions for Authors

I see great improvements and the article is now publishable. 

Comments on the Quality of English Language

Minor editing of English language required

Reviewer 2 Report

Comments and Suggestions for Authors

The authors addressed and responded appropriately to the questions and suggestions made.

Reviewer 4 Report

Comments and Suggestions for Authors

Good job.